# LEARNING THE KOOPMAN OPERATOR USING ATTENTION FREE TRANSFORMERS

## ABSTRACT

Learning Koopman operators with autoencoders enables linear prediction in a latent space, but long-horizon rollouts often drift off the learned manifold, leading to phase and amplitude errors on systems with switching, continuous spectra, or strong transients. We introduce two complementary components that make Koopman predictors substantially more robust. First, we add an *attention-free latent memory* (AFT) block that aggregates a short window of past latents to produce a corrective residual before each Koopman update. Unlike multi-head attention, AFT operates in linear time with nearly identical parameter count to the baseline, yet captures the local temporal context needed to suppress error divergence. Second, we propose *dynamic re-encoding*: lightweight, online change-point triggers (EWMA, CUSUM, and sequential two-sample tests) that detect latent drift and project predictions back onto the autoencoder manifold. Across three benchmark systems—Duffing oscillator, Repressilator, IRMA—our model consistently reduces error accumulation compared to a Koopman autoencoder and matched-capacity multi-head attention. We also compare against GRU and Transformer autoencoders, evaluated both from initial conditions and with a 50-step context, and find that Koopman+AFT (with optional re-encoding) attains markedly lower long-horizon error while maintaining substantially lower inference latency. We report improvements over horizons up to 1000 steps, together with ablations over trigger policies. The resulting predictors are fast, compact, and geometry-preserving, providing a practical path to long-term forecasting with Koopman methods.

## 1 INTRODUCTION

The Koopman operator offers a principled way to analyze nonlinear dynamics with linear tools by lifting states to an observable space where evolution is linear (Koopman, 1931). Neural implementations of this idea—most commonly, Koopman autoencoders (KAE) that learn an encoder $\varphi$, a linear map $K$, and a decoder $\varphi^{-1}$—often deliver strong single-step accuracy, but drift in long rolls: phases slip in oscillators, amplitudes decay or explode, and trajectories peel away from attractors (Lusch et al., 2018). Empirically, failures are pronounced in settings with (i) continuous or mixed spectra (e.g., undamped oscillators), (ii) switching between metastable basins, and (iii) transient regimes where small errors compound. This motivates mechanisms that (a) use short-term temporal context to correct local errors, akin to delay-embedding ideas in Hankel DMD / HAVOK (Arbabi & Mezic, 2017; Brunton et al., 2017), and (b) periodically project predictions back onto the learned manifold before drift becomes catastrophic. Robustness over long horizons is therefore critical: predictors that remain near the learned manifold accumulate fewer errors and are easier to certify and use in downstream control.

**Approach overview and intuition.** We augment a standard KAE with two complementary pieces. (i) An *attention-free latent memory* (AFT) block aggregates a short window of past latents and adds a corrective residual before each Koopman update, achieving linear time/memory in the context length while capturing the local correlations that drive phase and amplitude drift (Zhai et al., 2021). (ii) *Dynamic re-encoding* uses lightweight streaming triggers (EWMA, CUSUM, sequential two-sample, and simple threshold/window tests) to detect latent drift and apply an encode–decode–encode (E–D–E) projection that snaps predictions back to the autoencoder manifold (Roberts, 2000; Moustakides, 1986; Ross & Adams, 2012). Intuitively, AFT addresses *how* we step—reducing local error before

propagation by $K$—while re-encoding addresses *where* we step—bounding accumulated drift. The mechanisms are orthogonal: one *prevents* growth, the other *bounds* it.

**Relation to prior work.** Our approach builds on data-driven Koopman learning from EDMD with fixed dictionaries to learned latent embeddings with linearly recurrent bottlenecks (Li et al., 2017; Otto & Rowley, 2019; Lusch et al., 2018). Short time-delay context has long been used to stabilize prediction (Hankel DMD, HAVOK) (Arbabi & Mezic, 2017; Brunton et al., 2017), motivating our lightweight latent memory. Compared to transformer-style attention used in recent hybrids (Lu et al., 2024; Wang et al., 2022), our attention-free block achieves linear cost while targeting the local correlations that drive phase/amplitude drift. Orthogonally, projection/consistency ideas (Nayak et al., 2025; Frion et al., 2025; Noack et al., 2015; Dylewsky et al., 2019; Guan et al., 2024) inspire our encode–decode–encode snap-back mechanism. For triggering, we adopt classic streaming drift detectors (EWMA, CUSUM, sequential two-sample) (Roberts, 2000; Moustakides, 1986; Ross & Adams, 2012). Broader lines on inputs and control (KIC, Koopman MPC, safety/verification) are discussed in Appendix B, along with domain-specific applications in biology and fluid mechanics.

**Benchmarks.** We target three representative systems that stress long-horizon stability in complementary ways: (i) the **Duffing oscillator** in the unforced, undamped regime, which exhibits closed orbits, switching between wells at higher energies, and commonly a continuous or mixed Koopman spectrum that stresses linear predictors (Otto & Rowley, 2019; Li et al., 2017; Pan & Duraisamy, 2020; Alford-Lago et al., 2022; Köhne et al., 2025); (ii) the **Repressilator**, a synthetic three-gene negative-feedback oscillator with a canonical limit cycle (Elowitz & Leibler, 2000), widely used to evaluate identification and control (Boddupalli et al., 2019; Sootla et al., 2018; Balakrishnan et al., 2022; Perez-Carrasco et al., 2018); and (iii) **IRMA** (*In vivo Reverse-engineering and Modelling Assessment*), a five-gene yeast circuit constructed as a benchmark for modeling and control (Cantone et al., 2009; Marucci et al., 2009; Menolascina et al., 2014; di Bernardo et al., 2011) and representative of multi-gene regulatory dynamics where deep Koopman approaches have shown promise (Hasnain et al., 2019). These three cover, respectively, mixed spectra and switching (Duffing), clean oscillatory behavior with phase sensitivity (Repressilator), and higher-dimensional regulatory dynamics with intertwined feedback (IRMA).

**Empirical summary.** We evaluate on the three primary benchmarks (Duffing, Repressilator, IRMA) and report both MSE and a long-horizon *mean cumulative absolute error* (MCAE) that is sensitive to error accumulation. The latent memory block outperforms matched-capacity MHA (4 and 10 heads) on these systems, and coupling it with dynamic re-encoding yields the most robust rollouts. GRU and Transformer autoencoders, evaluated both from initial conditions and with a 50-step context, underperform on long horizons despite their added context.

**Contributions.**

- **Attention-free latent memory for Koopman prediction.** A linear-time, low-overhead block (Zhai et al., 2021) aggregates a short history of latents to produce a corrective residual before each Koopman update, substantially reducing error accumulation on long rollouts.
- **Dynamic re-encoding via streaming change detection.** An encode–decode–encode projection with online triggers (EWMA, CUSUM, sequential two-sample, threshold/window) detects latent drift and snaps predictions back to the learned manifold (Roberts, 2000; Moustakides, 1986; Ross & Adams, 2012).
- **Comprehensive evaluation and ablations on three representative systems.** On Duffing (unforced, undamped), Repressilator, and IRMA, latent memory outperforms matched MHA; latent memory + re-encoding attains the lowest MSE over 200/500/1000-step horizons; and gains persist across Koopman operator sizes.

## 2 METHODS

### 2.1 BASELINE KOOPMAN AUTOENCODER (KAE)

**Model.** Let $x_t \in \mathbb{R}^p$ denote the observed state at time $t$ and let $\varphi : \mathbb{R}^p \to \mathbb{R}^d$ and $\varphi^{-1} : \mathbb{R}^d \to \mathbb{R}^p$ be an encoder/decoder pair that maps to a $d$–dimensional latent space. The KAE posits a *linear*

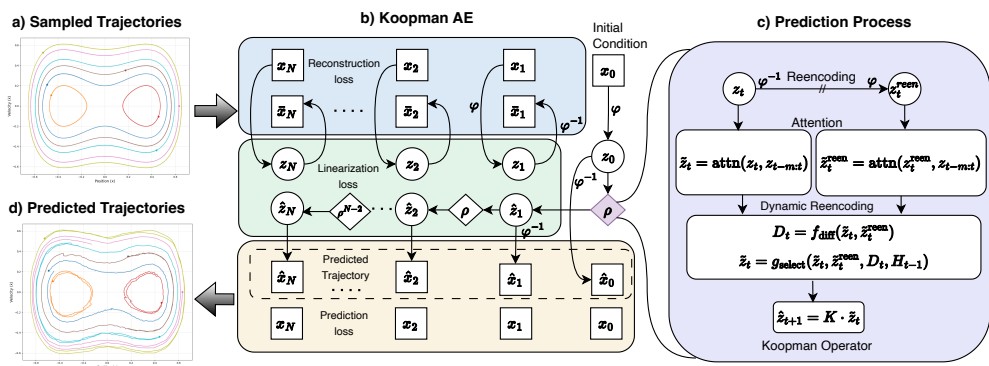

Figure 1: **Workflow of the Koopman autoencoder with AFT and Dynamic Re-encoding.** **(a)** Sampled trajectories from a Duffing Oscillator serve as input. **(b)** The core **Koopman autoencoder** learns a linear latent representation by minimizing reconstruction, linearization, and prediction losses. **(c)** The prediction process uses a **Dynamic Re-encoding** module with **AFT attention** to refine the latent state ($z_t \rightarrow \tilde{z}_t$), which is then evolved by the learned **Koopman operator K**. (d) The final output shows predicted trajectories that accurately replicate the system's dynamics.

latent evolution governed by a learned Koopman matrix $K \in \mathbb{R}^{d \times d}$:

$$z_t = \varphi(x_t), \qquad z_{t+1} = K z_t, \qquad \hat{x}_t = \varphi^{-1}(z_t), \tag{1}$$

so that an $i$–step rollout from an initial latent $y_0$ is $z_i = K^i z_0$ with decoded prediction $\hat{x}_i = \varphi^{-1}(K^i \varphi(x_0))$. We use the standard linearly recurrent bottleneck architecture (Otto & Rowley, 2019; Lusch et al., 2018) and learn $(\varphi, \varphi^{-1}, K)$ end-to-end.

**Training losses.** Given an input segment $(x_0, \ldots, x_T)$, we minimize a weighted sum of (i) reconstruction error, (ii) linearity consistency in the latent space, (iii) decoded prediction error over the rollout, and (iv) a unitary regularizer on $K$ to discourage exploding/vanishing spectra (cf. Enyeart & Lin, 2024):

$$\mathcal{L} = \alpha_1(\mathcal{L}_{\text{recon}} + \mathcal{L}_{\text{pred}}) + \mathcal{L}_{\text{lin}} + \alpha_2 \mathcal{L}_{\text{unitary}}, \tag{2a}$$

$$\mathcal{L}_{\text{recon}} = \frac{1}{T+1} \sum_{t=0}^{T} \|x_t - \varphi^{-1}(\varphi(x_t))\|_2^2, \tag{2b}$$

$$\mathcal{L}_{\text{lin}} = \frac{1}{T} \sum_{i=1}^{T} \|\varphi(x_i) - K^i \varphi(x_0)\|_2^2, \tag{2c}$$

$$\mathcal{L}_{\text{pred}} = \frac{1}{T} \sum_{i=1}^{T} \|x_i - \varphi^{-1}(K^i \varphi(x_0))\|_2^2, \tag{2d}$$

$$\mathcal{L}_{\text{unitary}} = \|KK^\top - I\|_F. \tag{2e}$$

with weights $\alpha_1, \alpha_2 > 0$. $\mathcal{L}_{\text{recon}}$ enforces an information-preserving autoencoding, $\mathcal{L}_{\text{lin}}$ encourages consistency of the latent trajectory with powers of $K$, and $\mathcal{L}_{\text{pred}}$ measures decoded multi-step accuracy. The unitary penalty mildly biases $K$ toward near-orthogonality to improve long-horizon stability (Enyeart & Lin, 2024). We train by unrolling equation 1 for $T$ steps from $x_0$, computing all four losses on the same segment. The formulation in equation 2 matches common KAE practice (Otto & Rowley, 2019; Lusch et al., 2018) while making the stability prior explicit.

## 2.2 Attention-Free Latent Memory (AFT)

**Setup.** To mitigate local phase/amplitude drift, we augment the KAE with a lightweight latent memory that aggregates the last $T$ latents before each Koopman step (here, $T$ denotes the *AFT*

*context length*, not the training segment length in equation 2). Let the latent history at time $t$ be $H_t = [z_{t-T}, \ldots, z_{t-1}] \in \mathbb{R}^{T \times d}$ (we use causal indexing and $T \ll$ rollout length). The AFT block produces a residual $\Delta z_t$ from $H_t$ and updates

$$\tilde{z}_{t-1} \;=\; z_{t-1} + \Delta z_t, \qquad z_t \;=\; K\,\tilde{z}_{t-1}, \tag{3}$$

so the Koopman propagator advances a *corrected* latent.

**Computation.** This variant is a plug-in replacement for Multi-Head Attention (MHA) and can be considered an element-wise linear attention mechanism. We use the AFT-full variant of this approach introduced by Zhai et al. (2021), where given a latent representation of $x_t$ in the Koopman subspace $Z_t$. We apply learned linear maps $W_Q, W_K, W_V \in \mathbb{R}^{d \times d}$,

$$Q_t \;=\; z_t W_Q \in \mathbb{R}^d, \qquad K_t \;=\; H_t W_K \in \mathbb{R}^{T \times d}, \qquad V_t \;=\; H_t W_V \in \mathbb{R}^{T \times d}, \tag{4}$$

then performs the following operation:

$$\Delta Z = f(Z); \quad \Delta z_t = \sigma_q(Q_t) \odot \frac{\sum_{t'=1}^T \exp(K_{t'} + w_{t,t'}) \odot V_{t'}}{\sum_{t'=1}^T \exp(K_{t'} + w_{t,t'})} \tag{5}$$

where $\sigma_q(\cdot)$ represents sigmoid activation on queries, $w \in \mathbb{R}^{T \times T}$ denotes learnable positional biases, and $\odot$ indicates element-wise multiplication. The mechanism computes attention weights from keys and positional biases, applies them to values, and then combines the result with activated queries through element-wise operations. This provides the benefits of attention mechanisms while maintaining linear computational complexity with respect to sequence length. In addition to that, we also added a key/value scaling by $\sqrt{d_{model}}$ to ensure numerical stability.

**Use in the predictor.** We apply equation 5 at every step with a rolling window $H_t$ (see Algorithm 1 in Appendix F for the full predictor). The block is *drop-in*: it changes neither the decoder nor the Koopman loss structure in equation 2. Empirically, the residual $\Delta z_t$ corrects local phase and amplitude errors before propagation by $K$, reducing error accumulation over long rollouts while preserving speed and model compactness.

## 2.3 DYNAMIC RE-ENCODING (E–D–E PROJECTION AND STREAMING TRIGGERS)

**Encode–Decode–Encode projection.** Let $\mathcal{P}(z) := \varphi(\varphi^{-1}(z))$ denote the autoencoder-induced projection of a latent $z$ back onto the learned manifold (idempotent by construction). During rollout, at each step, we form a pre-update latent $\tilde{z}_{t-1}$. We compute two one-step predictions:

$$z_t^{\text{pred}} \;=\; K\,\tilde{z}_{t-1}, \qquad z_t^{\text{re-pred}} \;=\; K\,\mathcal{P}(\tilde{z}_{t-1}).$$

Their discrepancy defines a *drift proxy*

$$\delta_t \;\triangleq\; \big\| z_t^{\text{re-pred}} - z_t^{\text{pred}} \big\|_2^2, \tag{6}$$

which grows when the iterate leaves the learned manifold. If a streaming trigger (below) fires at time $t$, we *snap* the latent back by replacing $z_{t-1} \leftarrow \mathcal{P}(\tilde{z}_{t-1})$ before propagating. This keeps the Koopman update on-manifold while leaving $K$ and the autoencoder unchanged.

**Streaming drift triggers.** We instantiate four inexpensive, streaming tests on the scalar $\{\delta_t\}$:

1. **Windowed Z-score (mean+std)**: maintain $\mu_t, \sigma_t$ over a sliding window of size $w$ and trigger if $\delta_t > \mu_t + \tau\,\sigma_t$ (hyperparameter $\tau > 0$).

2. **EWMA** (Roberts, 2000): update $Z_t = (1-\lambda)Z_{t-1} + \lambda\,\delta_t$ with $Z_0 = \delta_1$ and trigger when $|Z_t - \mu_Z| > L\,\sigma_Z$ (streaming estimates for $\mu_Z, \sigma_Z$; hyperparameters $\lambda \in (0,1), L > 0$).

3. **CUSUM** (Moustakides, 1986): compute the standardized cumulative sum $\tilde{s}_t = \frac{\sum_{i=1}^t (\delta_i - \mu_0)}{\sqrt{t}\,\sigma}$ and derive the p-value $p_t = 2\left[1 - \Phi\left(|\tilde{s}_t|\right)\right]$, where $\Phi$ is the standard normal CDF.

4. **Sequential two-sample** (Ross & Adams, 2012): compare a reference buffer $R$ and a current buffer $C$ (disjoint, size $w$) using a nonparametric test (e.g., KS or Lepage); re-encode if $p_t < \alpha$ (hyperparameter $\alpha$).

All tests have low computational overhead per step: windowed Z-score is $\mathcal{O}(w)$ for window size $w$, EWMA is $\mathcal{O}(1)$, and CUSUM is $\mathcal{O}(1)$ for time step $t$ with incremental updates. They are complementary: windowed Z-score/EWMA react quickly to level shifts, CUSUM accumulates small persistent deviations, and two-sample tests capture broader distributional changes. In our experiments, we use fixed hyperparameters per system and evaluate several trigger families (see Appendix 4.3). We use triggers only at *inference*; training proceeds without re-encoding.

## 3 EXPERIMENTS

### 3.1 BENCHMARKS AND DATA GENERATION

We evaluate three primary systems that emphasize long-horizon stability in complementary ways: (i) the **Duffing oscillator**, (ii) the **Repressilator**, and (iii) **IRMA** (introduced above). Further details about these systems are provided in Appendix A.1. To assess generality without excessive tuning, we additionally report results on a nonlinear pendulum, Goodwin oscillator, Lotka–Volterra, Rössler, and a reduced-order fluid-flow model (Goodwin, 1965; Fathi et al., 2023; Rössler, 1976; Noack et al., 2003). These are *sanity checks* performed with the finalized architecture to test out-of-the-box behavior; unlike the core trio, we did not perform extensive ablations or per-system tuning. Full details and additional figures are provided in the in the Appendix G.4. Parameterizations, time steps, numbers of trajectories, and training/prediction horizons follow Table 9 (data splits and any deviations are detailed in the Appendix). Full ODEs, solvers, parameter values, and initial-condition ranges for all systems are provided in Appendix H.

### 3.2 PROTOCOLS AND METRICS

**Rollout protocol.** Unless stated otherwise, models are trained on fixed-length segments and evaluated by free (open-loop) rollouts from test-set initial conditions. We report errors at horizons $\{200, 500, 1000\}$ steps on the three primary systems, and 200-step errors on the additional benchmarks.

**Metrics.** We report mean-squared error (MSE) at a fixed horizon and a long-horizon *mean cumulative absolute error* (MCAE) that captures accumulation of deviations. Given a rollout of length $H$, MCAE averages, across trajectories and state dimensions, the cumulative absolute error curve:

$$\text{MCAE}_t = \frac{1}{d} \sum_{j=1}^{d} \sum_{k=1}^{t} |\hat{x}_{k,j} - x_{k,j}| \tag{7}$$

$$\text{MCAE}_{\text{overall}} = \frac{1}{H} \sum_{t=1}^{H} \text{MCAE}_t \tag{8}$$

We plot MCAE over steps to reveal error growth dynamics.

**Hyperparameters & selection.** We fix the autoencoder bottleneck dimension and AFT context ($d{=}100$, $T{=}10$) across systems unless otherwise noted, and select early stopping and trigger thresholds on the validation set. The Koopman operator is dense by default.

### 3.3 BASELINES AND ABLATIONS

We compare:

1. **GRU**: the baseline GRU autoencoder (§E.1).
2. **Transformer**: the baseline transformer autoencoder (§E.2)
3. **KAE**: the baseline Koopman autoencoder (§2.1).
4. **KAE + AFT**: our latent-memory augmentation (§2.2).
5. **KAE + MHA**: matched-capacity multi-head attention with 4 or 10 heads (same bottleneck $d$, similar projection sizes).
6. **KAE + AFT + Re-enc**: dynamic re-encoding with streaming triggers (§2.3). We evaluate the sequential two-sample tests as the Dynamic Re-encoding Method using per-system validation-tuned thresholds, alongside periodic re-encoding from Fathi et al. (2023).

Ablations vary (i) the Koopman operator size, (ii) the AFT context $T$, and (iii) the trigger family/thresholds. For fairness, all baselines share the same autoencoder structure and training schedule.

## 4 RESULTS

### 4.1 PRIMARY COMPARISON ON THREE REPRESENTATIVE SYSTEMS

We conducted a comprehensive testing of our three primary systems for long-term horizon prediction. Our evaluation encompasses the reference models mentioned in §3.3. Additionally, we assessed GRU and Transformer architectures under two experimental conditions. Given that GRU and Transformer models require contextual information, we evaluated them first using only initial conditions, and then subsequently with a context of 50 time steps, which means in the Repressilator and IRMA escaping a large part of the transient state. Results are shown in Table 1.

**Duffing Oscillator.** Dynamic re-encoding is best at 200/500 steps (MSE 0.0113/0.0960), improving on AFT (0.0427/0.1536) and periodic re-encoding (0.0156/0.1187). At 1000 steps, AFT slightly leads (0.1947 vs. 0.2019), consistent with small snap-back–induced phase shifts accumulating over very long horizons. The vanilla KAE drifts (0.1286/0.2245/0.2471), and GRU/Transformer benefit from context yet remain far off Koopman variants (e.g., GRU 0.0862 vs. AFT 0.0427 at 200 steps). *Timely snap-backs help at switching transitions; for very long horizons, a small causal memory (AFT) often suffices.*

**Repressilator.** All Koopman variants handle the limit cycle, but AFT is decisively best across horizons (0.0001/0.0002/0.0005). Dynamic/periodic re-encoding degrade to $\sim\!4\!\times\!10^{-3}$ by injecting unnecessary phase resets. KAE is competitive at 200 steps (0.0002) but worsens by 1000 (0.0077). GRU/Transformer improve with context (e.g., GRU 0.0019 at 200) yet remain 10–100× worse than AFT. *On smooth limit cycles, prefer AFT-only; snap-backs are rarely needed and can be harmful.*

**IRMA.** Dynamic re-encoding is strongest and most stable (0.0001/0.0001/0.0003), with periodic close behind (0.0002/0.0004/0.0008). AFT is very good at short horizons (0.0004) but continues to degrade by the same rate (0.0009/0.0012). KAE collapses (10.1847 at 1000). GRU with context is competitive (0.0001/0.0003/0.0004) but from initial conditions is much worse (e.g., 0.0102 at 200). *GRU(+Ctx) benefits from being placed near the attractor; Koopman+AFT with snap-backs attains similar robustness without long input contexts.*

Table 1: Prediction performance comparison (MSE ↓) over different time steps across different system configurations. Best results for each system are highlighted in **bold**. Context provided for the GRU and Transformer is 50 time steps, while other results are from initial conditions, indicated as +Ctx and Init respectively.

| Steps | GRU | | Transformer | | Koopman | Koopman | AFT+Re-encoding | |
|---|---|---|---|---|---|---|---|---|
| | Init | +Ctx | Init | +Ctx | AE | AFT | Dynamic | Periodic |
| **Duffing Oscillator** | | | | | | | | |
| 200 | 0.2677 | 0.0862 | 0.2467 | 0.1868 | 0.1286 | 0.0427 | **0.0113** | 0.0156 |
| 500 | 0.2641 | 0.1981 | 0.3037 | 0.2441 | 0.2245 | 0.1536 | **0.0960** | 0.1187 |
| 1000 | 0.2556 | 0.2210 | 0.3196 | 0.2510 | 0.2471 | **0.1947** | 0.2019 | 0.2203 |
| **Repressilator** | | | | | | | | |
| 200 | 0.0081 | 0.0019 | 0.0079 | 0.0035 | 0.0002 | **0.0001** | 0.0041 | 0.0042 |
| 500 | 0.0090 | 0.0073 | 0.0193 | 0.0061 | 0.0028 | **0.0002** | 0.0041 | 0.0035 |
| 1000 | 0.0098 | 0.0092 | 0.0269 | 0.0085 | 0.0077 | **0.0005** | 0.0062 | 0.0067 |
| **IRMA** | | | | | | | | |
| 200 | 0.0102 | 0.0001 | 0.0109 | 0.0091 | 0.0171 | 0.0004 | **0.0001** | 0.0002 |
| 500 | 0.0076 | 0.0003 | 0.0404 | 0.0202 | 0.0935 | 0.0009 | **0.0001** | 0.0004 |
| 1000 | 0.0044 | 0.0004 | 0.0495 | 0.0251 | 10.1847 | 0.0012 | **0.0003** | 0.0008 |

### 4.2 ATTENTION VS AFT COMPARISON

We compare the latent-memory augmentation (AFT) to matched-capacity multi-head attention (MHA; 4 and 10 heads) on Duffing, Repressilator, and IRMA using both MSE and long-horizon

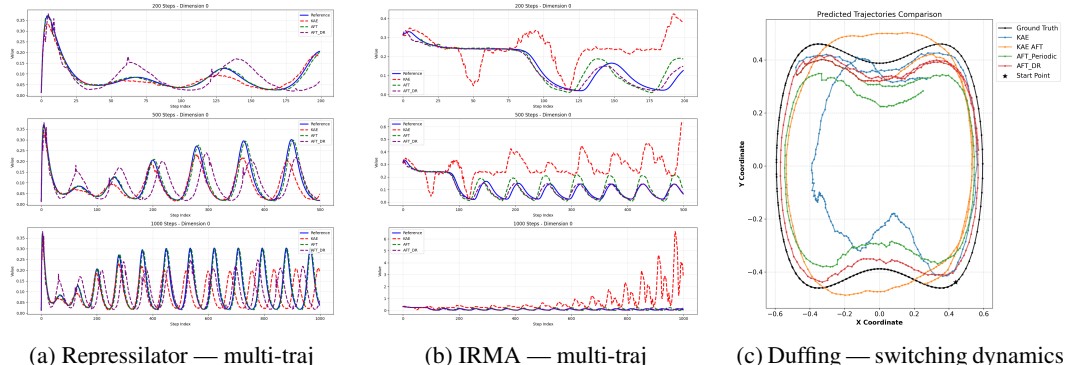

(a) Repressilator — multi-traj  (b) IRMA — multi-traj  (c) Duffing — switching dynamics

Figure 2: **Multi-trajectory rollouts on dynamical systems.** AFT reduces phase drift across initial conditions and enables accurate detection of switching dynamics in bistable systems.

MCAE. Figure 3 shows representative trajectories (top) and MCAE curves (bottom); summary metrics appear in Table 2.

**Duffing Oscillator.** AFT achieves the lowest error by a wide margin (MSE 0.0124 vs. 0.0957 / 0.1137 for 10/4-head MHA; $\sim 8$–$9\times$ lower), and flattens error growth (MCAE 10.95 vs. 49.09 / 52.98; $\sim 4.5$–$4.8\times$ lower). This matches the intuition that short, causal context suppresses phase/amplitude drift induced by the mixed/continuous spectrum and switching dynamics better than quadratic-cost attention.

**Repressilator.** On the clean limit cycle, AFT again dominates (MSE $3 \times 10^{-4}$ vs. $1.6 \times 10^{-3}$ / $1.8 \times 10^{-3}$; $\sim 5$–$6\times$ lower). MCAE is likewise reduced (1.80 vs. 5.26 / 5.66; $\sim 3\times$). The small, causal window corrects local misalignments before they accumulate into phase slips, yielding smoother, phase-consistent rollouts than MHA.

**IRMA.** AFT yields the best single-model accuracy (MSE $1 \times 10^{-4}$ vs. $1.2 \times 10^{-3}$ / $1.5 \times 10^{-3}$; $\sim 12$–$15\times$). MCAE also favors AFT (0.98 vs. 4.54 / 4.90; $\sim 4.6$–$5.0\times$), but the remaining long-horizon drift motivates using *AFT + re-encoding* (Sec. 4.3) on this higher-dimensional, feedback-rich system.

Overall, AFT consistently outperforms matched-capacity MHA across systems and metrics while retaining *linear* cost in the context length (cf. Sec. G.1), making it both more accurate and more scalable for long-horizon prediction.

Table 2: Performance comparison of attention mechanisms across different dynamical systems. Lower MSE and CMAE values indicate better performance. Best results are highlighted in bold.

| Model | MSE ($\downarrow$) | | | MCAE ($\downarrow$) | | |
|---|---|---|---|---|---|---|
| | 4MHA | 10MHA | AFT | 4MHA | 10MHA | AFT |
| Duffing Oscillator | 0.1137 | 0.0957 | **0.0124** | 52.9835 | 49.0874 | **10.9522** |
| Repressilator | 0.0018 | 0.0016 | **0.0003** | 5.6606 | 5.2564 | **1.7998** |
| IRMA | 0.0012 | 0.0015 | **0.0001** | 4.9036 | 4.5401 | **0.9786** |

## 4.3 EFFECT OF DYNAMIC RE-ENCODING (STREAMING TRIGGERS)

We study the dynamic re-encoding with streaming triggers on the Duffing oscillator (§2.3). Table 3 shows that the sequential two-sample detector attains the lowest error (0.0113), followed by EWMA and CUSUM, and Fig. 4 shows the sensitivity and accuracy of the methods. The two-sample method relies on statistical distribution changes, which account for better detection, whereas EWMA/CUSUM uses aggregated statistics that can smooth over subtle but meaningful shifts. Threshold-based methods are prone to false positives/negatives, as they do not fully account for prediction memory, and drift proxies incorporate not only manifold distance but also recon-

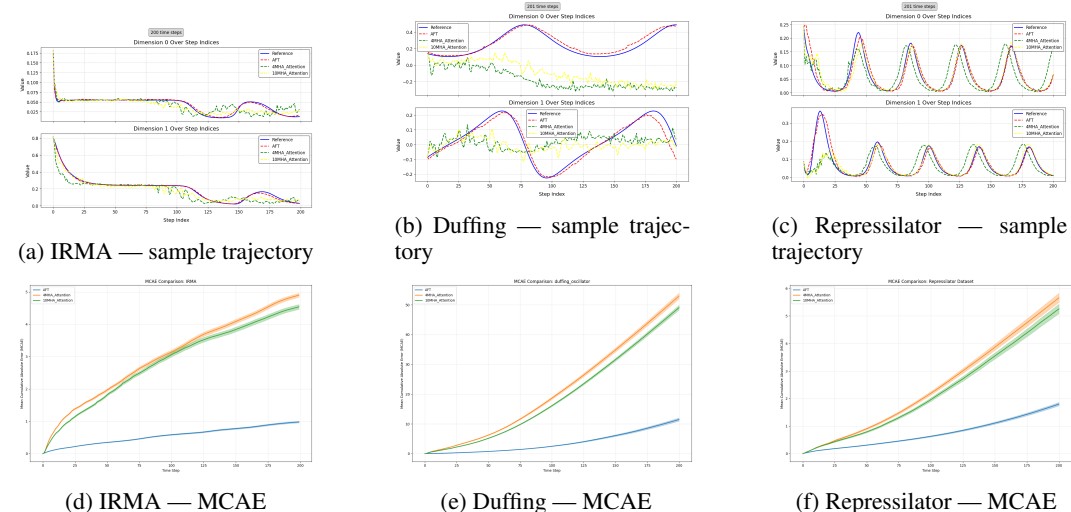

(a) IRMA — sample trajectory

(b) Duffing — sample trajectory

(c) Repressilator — sample trajectory

(d) IRMA — MCAE

(e) Duffing — MCAE

(f) Repressilator — MCAE

Figure 3: **AFT vs. MHA (4 and 10 heads) on the three primary systems.** The top row shows sampled trajectories, and the bottom row shows the MCAE curves. AFT reduces error growth and outperforms matched-capacity MHA.

struction error. While periodic re-encoding can achieve good performance, it primarily targets drift frequency and does not consider when or where the change occurs, limiting its responsiveness.

Table 3: Overall MSE per trajectory for different re-encoding methods.

| Method | AFT | Threshold | Window | Periodic | CUSUM | EWMA | TwoSample |
|---|---|---|---|---|---|---|---|
| **MSE** | 0.0427 | 0.0290 | 0.0186 | 0.0156 | 0.0151 | 0.0144 | **0.0113** |

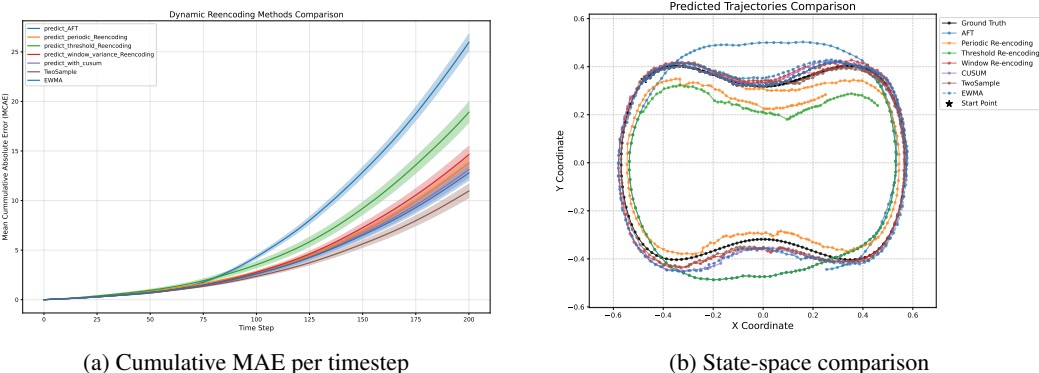

(a) Cumulative MAE per timestep

(b) State-space comparison

Figure 4: Comparison of re-encoding methods. (a) Cumulative MAE per timestep comparison across methods. (b) Comparison in latent state space for a sampled trajectory.

## 5 DISCUSSION

**What the latent memory buys.** Across Duffing, Repressilator, and IRMA, the attention-free latent memory (§2.2) consistently reduces phase slippage and amplitude drift over long horizons relative to the plain KAE and to a matched-capacity multi-head attention (MHA) baseline, lowering both MCAE and MSE (see §4.2, Table 2, Fig. 3; and §4.1, Table 1). Empirically, a short, causal context ($T=10$; §G.5) is sufficient to capture the local temporal correlations that most affect error accumulation, and doing so in $\mathcal{O}(Td)$ time/memory per step yields stable rollouts without the quadratic attention overhead (complexity details in App. G.1; context ablation in Fig. 8, right). This is a

pragmatic complement to spectral-accuracy pursuits in Koopman learning (Korda & Mezić, 2018; Mezić, 2022; Giannakis & Valva, 2024; Colbrook & Townsend, 2024): even when the learned $K$ is an imperfect global surrogate, local residual correction can substantially improve long-horizon behavior. Conceptually, the AFT residual acts like a short learned delay-embedding/HAVOK-style forcing term (Arbabi & Mezic, 2017; Brunton et al., 2017), while re-encoding is a latent-space analogue of windowed/recursive DMD projections (Noack et al., 2015; Guan et al., 2024).

**When and why re-encoding helps.** The dynamic re-encoding mechanism (§2.3) improves robustness primarily on systems with switching or stiff transients (e.g., Duffing at higher energies) or higher-dimensional, intertwined feedback (IRMA), where drifting off the autoencoder manifold can be abrupt and compounding. Quantitatively, triggers based on sequential two-sample tests attain the lowest MSE on Duffing, followed by EWMA and CUSUM (Table 3; Fig. 4; see also §4.3 and Alg. 2 in App. F). On clean, phase-sensitive oscillators (e.g., Repressilator), re-encoding can occasionally *hurt* when a trigger fires near a delicate phase region: the E–D–E projection introduces a small phase shift that the Koopman update then propagates (Table 1). Practical guidance: AFT-only for smooth limit cycles; AFT+EWMA/CUSUM for intermittent regime changes; and two-sample tests when residual distributions clearly separate nominal vs. drifted behavior.

**Sensitivity and hyperparameters.** Performance is most sensitive to (i) the quality of the autoencoder manifold, (ii) the AFT context $T$, and (iii) trigger thresholds. Too large a $T$ brings diminishing returns and mild over-smoothing (Fig. 8, right). Thresholds selected on validation data transfer well across test horizons in our runs, but overly aggressive settings can over-trigger and degrade smooth oscillations. A dense $K$ offered the strongest accuracy (consistent with prior observations), whereas structured variants (diagonal, banded, Jordan) trade accuracy for interpretability; we include these ablations for completeness (Fig. 8, left).

**Applications and impact.** Where long-horizon forecasting is needed under tight computational budgets (embedded monitoring, rapid what-if simulation), the $\mathcal{O}(Td)$ latent memory and occasional E–D–E snaps provide a practical path that keeps the standard KAE backbone intact and reproducible. Compared to GRU and Transformer autoencoders (architectures in App. E), Koopman-based predictors deliver both stronger long-horizon fidelity (Table 1) and substantially lower latency (Table 5). Breadth checks across additional dynamical systems indicate out-of-the-box gains where appropriate (App. G.4; Table 7, Fig. 7).

# 6 LIMITATIONS AND FUTURE WORK

Our stability claims are empirical: we do not provide convergence or spectral-error guarantees for the learned $K$ despite relevant theory (Korda & Mezić, 2018; Mezić, 2022; Giannakis & Valva, 2024; Colbrook & Townsend, 2024). Effectiveness depends on the autoencoder manifold; if $\varphi^{-1}$ is lossy, the E–D–E projection can bias latents. Trigger policies introduce hyperparameters (thresholds, windows) and can degrade performance on clean limit cycles (Table 1, Repressilator) even while helping on systems with switching or stiff transients (Duffing, IRMA; Tables 1, 3, Fig. 4). Dynamic re-encoding is used only at inference, so the model is not co-trained with snaps. Some configurations still degrade at very long horizons (e.g., AFT on IRMA at 1000 steps in Table 1), and vanilla KAE can collapse. Our experiments focus on autonomous systems; inputs/control are out of scope here. Additional benchmarks suggest "out-of-the-box" generality, but we did not target per-system SOTA.

Future directions include bridging empirical robustness with guarantees (resolvent/residual-minimization objectives and stability-biased constraints for $K$; EDMD diagnostics during training (Giannakis & Valva, 2024; Colbrook & Townsend, 2024; Mezić, 2022; Korda & Mezić, 2018)), training curricula that transition from one-step to free rollouts, uncertainty-aware or learned triggers that retain the two-sample sensitivity benefits on Duffing (Table 3) while avoiding false snaps on smooth cycles, and adaptive memory that learns/gates the AFT context $T$ (cf. Fig. 8). Extending AFT and re-encoding to controlled settings (DMDc/EDMDc/KIC) and evaluating in receding-horizon MPC is natural, as is studying partial/noisy/hybrid systems. Finally, exploring structured $K$ for interpretability with minimal loss, and fusing AFT with decoders for hardware-efficient deployment, are promising for resource-constrained use (Table 5).

## 7 Reproducibility Statement

We have made every effort to ensure the reproducibility of our results. The paper provides detailed descriptions of the model architecture, training setup, and evaluation protocols. Hyperparameters, dataset generation, and experimental settings are included in the Appendix. We have also provided a detailed reproducibility checklist in the Appendix C, which outlines the entire experimental process step by step. To further support replication, we have made the code available at `https://anonymous.4open.science/r/Attended-Koopman-3E85` for review. We will release the final version publicly upon publication.

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

# A SYSTEMS AND DATA GENERATION

## A.1 DYNAMICAL SYSTEMS

### A.1.1 DUFFING OSCILLATOR

The Duffing oscillator represents a paradigmatic example of nonlinear dynamics described by the second-order differential equation

$$\ddot{x} + \delta\dot{x} + \alpha x + \beta x^3 = \gamma\cos(\omega t).$$

This system has been studied in the context of data-driven modelling and Koopman operator theory due to its rich dynamical behaviour and analytical tractability (Otto & Rowley, 2019; Li et al., 2017; Alford-Lago et al., 2022; Pan & Duraisamy, 2020; Köhne et al., 2025).

In this work, we focus on the unforced and undamped case governed by

$$\ddot{x} = x - x^3, \tag{9}$$

which captures essential features such as switching dynamics between stable states and (mixed/)continuous spectrum characteristics, making it ideal for showcasing our proposed methodology. The state-space representation is

$$\begin{aligned} \dot{x}_1 &= x_2, \\ \dot{x}_2 &= x_1 - x_1^3. \end{aligned} \tag{10}$$

Here $x_1$ denotes position and $x_2$ velocity. Initial conditions are sampled uniformly from $(x_1, x_2) \in [-2, 2]$ with fixed step size as in Table 9. This configuration admits two stable centers at $(x_1, x_2) = (\pm 1, 0)$ and an unstable fixed point at the origin $(0, 0)$. The system is Hamiltonian and trajectories form closed orbits in phase space: low-energy orbits are confined to individual potential wells, while high-energy orbits encircle both wells, exhibiting switching behavior as trajectories periodically transition between states.

The continuous-spectrum nature of this system poses significant challenges for traditional Koopman operator approximation methods, as there is no straightforward finite-dimensional approximation in terms of a small number of eigenfunctions. Additionally, the switching dynamics between potential wells create computational difficulties even for short-term prediction.

### A.1.2 REPRESSILATOR

The Repressilator, a popular synthetic gene circuit (Elowitz & Leibler, 2000), has become a canonical model for studying oscillatory dynamics that emerge from negative feedback regulation. The circuit is composed of three transcriptional repressor genes arranged in a cyclic negative feedback loop, where each gene encodes a protein that inhibits the transcription of the next gene in the cycle, creating a ring-like structure. Many studies extensively analysed data-driven modelling and control of such systems in (Boddupalli et al., 2019; Sootla et al., 2018; Balakrishnan et al., 2022; Perez-Carrasco et al., 2018). Here, we focus on the case where the genetic circuit is isolated from bacterial host (Weiße et al., 2015; Nikolados et al., 2019) and admits a limit cycle in the phase portrait with a single basin of attraction centered at the origin. We define the system as:

$$\begin{aligned} \frac{dm_{(i)}}{dt} &= -\delta_m m_{(i)} + \frac{\alpha}{1 + (p_{(j)}/K)^n} + \alpha_0, \\ \frac{dp_{(i)}}{dt} &= -\delta_p p_{(i)} + \beta m_{(i)}, \end{aligned} \tag{11}$$

where $m_i$ and $p_i$ denotes the concentration of mRNA and protein of gene $i$, respectively. The indices $(i, j)$ cycles through the repressor pairs $\{(\text{lacI}, \text{cI}), (\text{tetR}, \text{lacI}), (\text{cI}, \text{tetR})\}$. The model parameters represent basal and maximal transcription rates $(\alpha_0, \alpha)$, Hill repression characteristics $(K, n)$, degradation rates $(\delta_m, \delta_p)$, and the translation rate $(\beta)$. We used the parameter values $\alpha_0 = 0.03$, $\alpha = 10$, $K = 40$, $n = 2$, $\delta_m = 0.3466$, $\delta_p = 0.0693$, and $\beta = 10$ in dimensionless units.

### A.1.3 IRMA

The IRMA (In vivo Reverse-engineering and Modelling Assessment) network is a well-characterised synthetic gene circuit in *Saccharomyces cerevisiae*, constructed explicitly as a benchmark for modelling and control. IRMA consists of five yeast transcription-factor genes (CBF1, GAL4, SWI5, ASH1, GAL80) with a topology containing both positive and negative feedback loops (Marucci et al., 2009; di Bernardo et al., 2011). It was designed to be insulated from native regulation and to respond specifically when cells are cultured in galactose. This network has been used to test system-identification and control methods. For example, Menolascina et al. (2014) applied closed-loop control to regulate IRMA's reporter output, and Cantone et al. (2009) used IRMA time-series data to validate reverse-engineering algorithms. These studies demonstrate IRMA's predictive modelling value.

The mathematical model of IRMA is characterised by the following system of equations:

$$\dot{x_1} = \alpha_1 + v_1 \cdot \frac{k_1^{h_1}}{k_1^{h_1} + x_5^{h_1}} - d_1 x_1,$$

$$\dot{x_2} = \alpha_2 + v_2 \cdot \frac{x_1^{h_2}}{k_2^{h_2} + x_1^{h_2}} - d_2 x_2,$$

$$\dot{x_3} = \alpha_3 + v_3 \cdot \frac{x_2^{h_3}}{k_3^{h_3} + x_2^{h_3} \left(1 + \frac{x_4^{h_6}}{\gamma^{h_6}}\right)} - d_3 x_3, \tag{12}$$

$$\dot{x_4} = \alpha_4 + v_4 \cdot \frac{x_3^{h_4}}{k_4^{h_4} + x_3^{h_4}} - d_4 x_4,$$

$$\dot{x_5} = \alpha_5 + v_5 \cdot \frac{x_3^{h_5}}{k_5^{h_5} + x_3^{h_5}} - d_5 x_5.$$

where $x_1, x_2, x_3, x_4, x_5$ represent CBF1, GAL4, SWI5, GAL80, and ASH1 respectively, parameters follow the implementation in Marucci et al. (2009) and states are sampled from a uniform distribution over the interval $[0, 1]$.

The parameters of the model include the basal expression rates $\alpha_i$, the maximum expression rates $v_i$, the half-saturation constants $k_i$, the Hill coefficients $h_i$, the degradation rates $d_i$, and the inhibition constant $\gamma$. Together, these parameters govern the nonlinear gene regulatory interactions and degradation dynamics of the IRMA circuit.

## A.2 ADDITIONAL DYNAMICAL SYSTEMS

In addition to the three main benchmark systems (A.1, we also consider a collection of classical dynamical systems as benchmarks. These systems are commonly used in the literature for system identification tasks, as they display diverse and rich dynamical behaviors. We briefly describe each of them below.

**Nonlinear Pendulum.** The pendulum represents a freely swinging pole. Unlike the linear small-angle approximation, the full nonlinear pendulum exhibits richer dynamics. As the system energy increases, oscillations become strongly anharmonic, leading to a continuous Koopman spectrum. The dynamics are given by:

$$\dot{x_1} = x_2, \tag{13}$$
$$\dot{x_2} = -\sin(x_1). \tag{14}$$

Initial conditions with angular positions $\theta_0$ from a uniform distribution over $[-\pi, \pi]$ radians, with angular velocities fixed at $\omega_0 = 0.0$.

**Parabolic Attractor.** Adopted from Lusch et al. (2018), this simple dynamical system has a single fixed point and a discrete eigenvalue spectrum:

$$\dot{x_1} = \mu x_1, \tag{15}$$

$$\dot{x_2} = \lambda\big(x_2 - x_1^2\big). \tag{16}$$

The system exhibits a slow manifold for stable eigenvalues $\lambda < \mu < 0$, asymptotically attracted to the parabola $x_2 = x_1^2$. We set $\lambda = -1.0$ and $\mu = -0.1$, with initial conditions sampled uniformly from $x_1, x_2 \in [-1, 1]$.

**Goodwin Oscillator.** The three-state Goodwin oscillator (Goodwin, 1965) is a prototypical biochemical feedback model demonstrating how delayed negative feedback generates self-sustained oscillations. It consists of three variables (commonly interpreted as mRNA, protein, and inhibitor), where the inhibitor suppresses mRNA production. The system is governed by:

$$\dot{x_1} = \frac{\alpha}{\kappa + k x_3^n} - \beta x_1, \tag{17}$$

$$\dot{x_2} = \gamma x_1 - \delta x_2, \tag{18}$$

$$\dot{x_3} = \eta x_2 - \theta x_3, \tag{19}$$

where $(x_1, x_2, x_3)$ denote the concentrations of the three states and are sampled from uniform distributions over $[-2, 2]$ for each state variable and used the following parameters to generate oscillation: $a_1 = 360$, $\kappa_1 = 43$, $k_1 = 1.0$, $n = 12$, $b_1 = 0.6$, $\alpha_1 = 1.0$, $\beta_1 = 1.0$, $\gamma_1 = 1.0$, and $\delta_1 = 0.8$.

**Lotka–Volterra System.** The Lotka–Volterra equations describe a classical predator-prey model whose populations can undergo sustained oscillations:

$$\dot{x_1} = \alpha x_1 - \beta x_1 x_2, \tag{20}$$

$$\dot{x_2} = \delta x_1 x_2 - \gamma x_2. \tag{21}$$

The system admits two fixed points: extinction at $(0, 0)$ and coexistence at $\big(\frac{\gamma}{\delta}, \frac{\alpha}{\beta}\big)$. We follow the setup of Fathi et al. (2023) and set $\alpha = \beta = \gamma = \delta = 0.2$, with initial conditions sampled uniformly from $x_1, x_2 \in [0.02, 3.0]$.

**Rössler System.** The Rössler system (Rössler, 1976) is a three-dimensional chaotic system defined by:

$$\dot{x_1} = -x_2 - x_3, \tag{22}$$

$$\dot{x_2} = x_1 + a x_2, \tag{23}$$

$$\dot{x_3} = b + x_3(x_1 - c). \tag{24}$$

With the canonical parameter set $(a, b, c) = (0.2, 0.2, 5.7)$, the system yields the well-known strange attractor characterized by oscillations in the $(x_1, x_2)$-plane and intermittent growth/decay along $x_3$.

**Fluid Flow Model.** A reduced-order model of fluid flow past a circular cylinder at Reynolds number 100 (Noack et al., 2003) is given by:

$$\dot{x_1} = \mu x_1 - \omega x_2 + A x_1 x_3, \tag{25}$$

$$\dot{x_2} = \omega x_1 + \mu x_2 + A x_2 x_3, \tag{26}$$

$$\dot{x_3} = -\lambda\big(x_3 - x_1^2 - x_2^2\big). \tag{27}$$

With parameters $\mu = 0.1$, $\omega = 1.0$, $A = -0.1$, and $\lambda = 10$, this system serves as a benchmark for fluid dynamics, exhibiting self-sustained von Kármán vortex shedding. We consider trajectories starting both on and off the slow manifold.

## B  EXTENDED RELATED WORK

### B.1  KOOPMAN OPERATOR LEARNING

Data-driven approximations of the Koopman operator (Koopman, 1931) have matured from dictionary-based linear models to learned latent embeddings. Extended DMD (EDMD) introduces

a finite dictionary of observables and performs linear regression in the lifted space (Li et al., 2017). Rigorous analyses quantify when EDMD converges and how spectra are approximated (Korda & Mezić, 2018; Mezić, 2022; Giannakis & Valva, 2024; Colbrook & Townsend, 2024). Neural formulations replace hand-crafted dictionaries with encoders/decoders that learn Koopman-invariant coordinates end-to-end, often with a linearly recurrent bottleneck (Otto & Rowley, 2019; Lusch et al., 2018). These approaches have been used on canonical nonlinear systems—including Duffing-type oscillators—to demonstrate improved single-step prediction and limited-horizon rollout accuracy (Otto & Rowley, 2019; Li et al., 2017; Alford-Lago et al., 2022; Pan & Duraisamy, 2020; Köhne et al., 2025). Unlike black-box recurrent-based neural network models such as LSTM and GRU (Elman, 1990; Hochreiter & Schmidhuber, 1997; Cho et al., 2014), Koopman-based methods yield interpretable latent coordinates and preserve better system dynamics in extrapolation. Regularizers that bias the learned propagator toward (near-)unitary dynamics have been explored to stabilize long-horizon rollouts (Enyeart & Lin, 2024). Symmetry-aware variants study how equivariances shape Koopman spectra and model structure (Salova et al., 2019).

## B.2 Delay embeddings and memory

A parallel line of work augments Markov predictors with short-term memory via time-delay embeddings. Hankel DMD constructs a block-Hankel snapshot matrix to expose linear evolution in delay coordinates (Arbabi & Mezic, 2017); related theory develops universal, system-independent time-delay observables (Kamb et al., 2020). HAVOK (Hankel Alternative View of Koopman) further separates a low-dimensional linear model from a data-driven forcing term that captures intermittent or chaotic dynamics (Brunton et al., 2017). These methods show that short windows of history can substantially reduce phase slippage and amplitude drift, motivating lightweight latent-memory mechanisms in neural Koopman models.

## B.3 Attention and hybrid Koopman models

Recent models couple Koopman structure with attention to aggregate recent context or to adapt locally. Lu et al. (2024) employ temporal attention inside an autoencoder to attenuate noise and improve forecasting; Wang et al. (2022) pair a global (stationary) Koopman map with a local transformer-based operator to handle nonstationarity and transients. We follow the same spirit—leveraging short temporal context for robust prediction—while replacing quadratic-cost multi-head attention with a linear-cost, attention-free aggregation in latent space.

## B.4 Inputs, control, and MPC

Complementary work integrates inputs and control: KIC extends Koopman predictors to systems with inputs (Proctor et al., 2018); LNCIS surveys detail operator-learning pipelines for control (Kaiser et al., 2020); Koopman MPC demonstrates closed-loop planning in lifted coordinates (Korda & Mezić, 2020); and recent surveys emphasize applications in robot learning (Shi et al., 2024). These applications motivate robustness over long horizons, as models that remain near the learned manifold are easier to certify and use in downstream control.

## B.5 Projection, consistency, and recursive/local modeling

To limit compounding errors, several practices periodically project or reconcile predictions with the learned manifold. Temporal consistency regularization encourages smooth, self-consistent multi-step predictions (Nayak et al., 2025); delayed-input concatenation provides a simple memory buffer for low-dimensional series (Frion et al., 2025). In the classical setting, windowed/recursive DMD maintains local linear surrogates from sliding subsets of recent data (Noack et al., 2015; Dylewsky et al., 2019), and recent variants use windowed outputs to update local linear models online (Guan et al., 2024). We operationalize a complementary idea in latent space: a cheap encode–decode–encode projection that snaps predictions back to the autoencoder manifold when drift is detected.

### B.6 Streaming drift detection

Change-point detection from statistical process control offers streaming triggers that are inexpensive and interpretable. CUSUM tests cumulative deviations against a nominal mean (Moustakides, 1986); EWMA emphasizes recent residuals through exponential smoothing (Roberts, 2000); and sequential two-sample procedures compare reference and current windows to detect broader distributional shifts (Ross & Adams, 2012). We instantiate all three as latent-drift monitors to decide when to re-encode.

### B.7 Biological circuits and broader benchmarks

Synthetic gene networks furnish controlled, nonlinear testbeds with oscillations and feedback. The Repressilator (Elowitz & Leibler, 2000) and the IRMA network (*In vivo Reverse-engineering and Modelling Assessment*) (Marucci et al., 2009; di Bernardo et al., 2011) have been repeatedly used for modeling and closed-loop control (Menolascina et al., 2014; Cantone et al., 2009). Koopman-based predictors and controllers have also been explored for genetic circuits (Hasnain et al., 2019). Beyond biology, standard dynamical-systems benchmarks probe complementary difficulties: the Goodwin oscillator (Goodwin, 1965), Rössler attractor (Rössler, 1976), and reduced-order cylinder flow (Noack et al., 2003), as well as pedagogical systems such as the parabolic attractor (Lusch et al., 2018) and Lotka–Volterra (Fathi et al., 2023). In our experiments, we focus our most rigorous evaluation on three representative systems (Duffing, Repressilator, IRMA) and use the remaining benchmarks to sanity-check generalization of the finalized architecture.

## C Reproducibility Checklist

**Code and data.** We release code, configuration files, and scripts to (i) generate datasets for all systems, (ii) train/evaluate each model variant (KAE, KAE+AFT, KAE+MHA, KAE+AFT+Re-enc), and (iii) reproduce all tables/figures. Re-encoding triggers (EWMA, CUSUM, windowed Z-score, two-sample) are provided as modular components.

**Training and evaluation protocol.** We fix optimizer, learning-rate schedule, batch size, rollout horizons, and early-stopping criteria as in §H. Models are evaluated by free rollouts from held-out initial conditions; we report MSE and MCAE as defined in §3.2. All reported means are over multiple random initial-condition with $95\%$ CIs for MCAE curves.

**Hyperparameters.** Architecture and training hyperparameters are summarized in Tables 8 and 9. Unless otherwise noted, the bottleneck is $d=100$, AFT context $T=10$, and $K$ is dense. Any deviations are stated near the corresponding results.

**Determinism and versions.** We provide exact library versions (PyTorch, CUDA, numpy, scipy) and OS details. Where relevant, we disable non-deterministic CuDNN kernels.

Table 4: Checklist of key reproducibility items and where they are specified.

| Item | Status | Where |
| --- | --- | --- |
| Datasets & generation scripts | Provided | App. H, App. A |
| Train/eval protocols | Specified | §H |
| Architectures & losses | Specified | §2.1, §2.2, equation 2 |
| Re-encoding triggers | Specified | §2.3, Alg. 2 |
| Hyperparameters (per system) | Tabulated | Tables 8, 9 |
| Baselines & ablations | Enumerated | §3.3 |
| Metrics (MSE, MCAE) | Defined | §3.2 |
| Code | Provided | § 7 |

**Data generation.** All ODE systems were integrated with `scipy.integrate.odeint`, a wrapper of ODEPACK's LSODA solver that automatically detects stiffness and switches between a variable–order Adams method (non-stiff) and a BDF/Gear method (stiff), with adaptive internal step sizes and default error control (relative and absolute tolerances left at SciPy/LSODA defaults) following common practice in prior Koopman and system-identification studies (see §B). Solutions

were returned at user–specified sample times $t_0, \ldots, t_T$ (uniform linspace per dataset), so the reported $\Delta t$ in tables refers to output sampling, not the solver's internal step. We did not supply Jacobians or event functions; LSODA formed finite–difference Jacobians as needed. Initial conditions were sampled from the ranges stated in Appendix A and, for each system, we generate separate train/validation/test sets by sampling initial conditions.

## D  DYNAMIC RE-ENCODING METHODS

To implement dynamic re-encoding, we considered a set of online change-point detection methods that can identify shifts in the drift error and decide when re-encoding is beneficial. These approaches vary in complexity, from simple threshold-based rules to more sophisticated statistical tests, but they all share the goal of adapting the model to evolving data. Below, we provide a brief description of each method:

1. **Cumulative Sum (CUSUM)**: Originating from the work of Moustakides (1986), CUSUM is a sequential analysis technique that monitors cumulative deviations of observations from a target mean. We employ a probabilistic variant that standardizes the observed MSE difference between predictions with and without re-encoding, computes the cumulative sum, and converts it into a standard normal statistic. We then derive a p-value

$$ p_T = 2 \left[ 1 - \Phi\big( |\tilde{s}_T| \big) \right], $$

   where $\Phi$ denotes the standard normal CDF. This p-value quantifies the improbability of the observed cumulative deviation under the no-change hypothesis.

2. **Threshold Re-encoding**: We quantify the discrepancy between the original latent prediction $Y_{\text{pred}}$ and the re-encoded prediction $Y_{\text{pred-after}}$ using a normalized mean squared difference:

$$ \Delta = \frac{\|Y_{\text{pred}} - Y_{\text{pred-after}}\|^2}{\|Y_{\text{pred}}\|^2 + \epsilon}. $$

   A re-encode is triggered when $\Delta$ exceeds a predefined threshold.

3. **Window Re-encoding**: We track the MSE difference between the standard and re-encoded predictions in a fixed-size sliding window. Re-encoding is activated if the most recent MSE exceeds the window's mean plus a configurable multiple of its standard deviation, enabling adaptive response to abnormal fluctuations while balancing stability and efficiency.

4. **Exponentially Weighted Moving Average (EWMA)**: Introduced in Roberts (2000), the EWMA method computes a smoothed statistic that emphasizes recent observations. The update rule is

$$ Z_t = (1 - \lambda)\, Z_{t-1} + \lambda\, \delta_t, \quad \lambda \in (0, 1), $$

   Where a new observation is $\delta_t$, the smoothing parameter is $\lambda$. The method maintains running estimates of the mean $\mu_t$ and the standard deviation $\sigma_Z$ of the EWMA statistic. A change point is declared if

$$ \frac{|Z_t - \mu_t|}{L} > \sigma_Z, $$

   with $L$ being a sensitivity scaling factor.

5. **Sequential Two-Sample Test**: Extending the methods of Ross & Adams (2012), this approach partitions the data stream into a "reference" and a "current" window buffer and applies nonparametric tests (e.g., Kolmogorov–Smirnov, Lepage, Mann–Whitney) to detect distributional shifts beyond mean changes—such as variance or skewness deviations.

## E  GRU AND TRANSFORMER ARCHITECTURES

We use GRUs and transformers in an autoencoder architecture as a baseline to compare with the Koopman autoencoders. We chose these as baselines due to their ability to model the temporal and spatial dependence from the training data. We use simple model architectures to allow the model to be as expressive as possible to learn from the provided data. A description of each autoencoder is provided below.

### E.1 GRU AUTOENCODER

**Model.** Let $X_t = (x_t, x_t + 1, \ldots, x_{t+T-1}) \in \mathbb{R}^{p \times T}$ denote the observed window of length $T$ of states at time $t$ and let $E_{gru} : \mathbb{R}^p \to \mathbb{R}^d$ be an $n$-layer GRU encoder and $D_{mlp} : \mathbb{R}^d \to \mathbb{R}^p$ be a one-layer MLP decoder. Together, $E_{gru}$ maps the input window of length $T$ to a $d$-dimensional latent space and $D_{mlp}$ decodes back to a next-step prediction in $\mathbb{R}^p$:

$$\hat{x}_{t+T} = D_{mlp}(E_{gru}(X_t)) \tag{28}$$

so that an $i$–step rollout from an initial observed window $X_0$ is $i$ autoregressive applications of the autoencoder, shown in algorithm 3.

**Training losses.** Given an input window $X_0$ of length $T$, we minimize the autoregressive prediction error over a rollout of $T_{pred}$ steps. Let $f_{gru\_ar} : X_0 \to \hat{X}_T$ denote the algorithm described in 3 for a $T_{pred}$-step model rollout with context length $T$.

$$\mathcal{L} = \frac{1}{T_{pred}} \big\| X_T - f_{gru\_ar}(X_0) \big\|_2^2, \tag{29a}$$

We train by using this loss function over the training data.

### E.2 TRANSFORMER AUTOENCODER

The transformer autoencoder is almost identical to the GRU autoencoder described in E.1 except that the encoder is now an $n$-layer GRU followed by an $m$-layer transformer encoder. The GRU head is used to embed the input data to a higher dimension before being passed through the transformer.

**Model.** Let $X_t = (x_t, x_t + 1, \ldots, x_{t+T-1}) \in \mathbb{R}^{p \times T}$ denote the observed window of length $T$ of states at time $t$ and let $E_{tr} : \mathbb{R}^p \to \mathbb{R}^d$ be an $(n + m)$-layer transformer encoder with and $D_{mlp} : \mathbb{R}^d \to \mathbb{R}^p$ be a one-layer MLP decoder. Together, $E_{tr}$ maps the input window of length $T$ to a $d$-dimensional latent space and $D_{mlp}$ decodes back to a next-step prediction in $\mathbb{R}^p$:

$$\hat{x}_{t+T} = D_{mlp}(E_{tr}(X_t)) \tag{30}$$

so that an $i$–step rollout from an initial observed window $X_0$ is $i$ autoregressive applications of the autoencoder, shown in algorithm 3.

**Training losses.** Given an input window $X_0$ of length $T$, we minimize the autoregressive prediction error over a rollout of $T_{pred}$ steps. Let $f_{t\_ar} : X_0 \to \hat{X}_T$ denote the algorithm described in 3 for a $T_{pred}$-step model rollout with context length $T$.

$$\mathcal{L} = \frac{1}{T_{pred}} \big\| X_T - f_{t\_ar}(X_0) \big\|_2^2, \tag{31a}$$

We train by using this loss function over the training data.

## F  ALGORITHMS

---

**Algorithm 1:** AFT–Koopman rollout (no re-encoding)

---

**Input:** initial state $x_0$, horizon $T_{\text{pred}}$, context length $T$

**Output:** predicted states $\{\hat{x}_t\}_{t=0}^{T_{\text{pred}}}$

**Data:** encoder $\varphi$, decoder $\varphi^{-1}$, Koopman map $K$, AFT params $W_Q, W_K, W_V$, position bias $B$, causal mask $M$

1  $z_0 \leftarrow \varphi(x_0); \quad \hat{x}_0 \leftarrow x_0;$

2  **for** $t \leftarrow 1$ **to** $T_{\text{pred}}$ **do**

    *// Assemble latent history (causal, chronological, length T)*

3    build $H_{t-1} = [z_{t-T}, \dots, z_{t-1}]$ (truncate if $t < T$);

    *// AFT projections*

4    $q \leftarrow z_{t-1} W_Q; \quad K_t \leftarrow H_{t-1} W_K; \quad V_t \leftarrow H_{t-1} W_V;$

    *// Residual aggregation in latent space*

5    $\alpha_{i,j} \leftarrow \exp(k_j + W_{i,j}) \cdot M_{i,j}$ for all $i, j$;

6    $\Delta z_i \leftarrow q_i \odot \frac{\sum_j \alpha_{i,j} \odot v_j}{\sum_j \alpha_{i,j}}$ for all $i$;

    *// Koopman propagation + decode*

7    $zt \leftarrow K(zt - 1 + \Delta z_{t-1}); \quad \hat{x}_t \leftarrow \varphi^{-1}(z_t);$

8  **end**

9  **return** $\{\hat{x}_t\}_{t=0}^{T_{\text{pred}}}$

---

---

**Algorithm 2:** AFT–Koopman rollout with dynamic re-encoding (inference only)

---

**Input:** initial state $x_0$, horizon $T_{\text{pred}}$, context length $T$, trigger config $\Theta$

**Output:** predicted states $\{\hat{x}_t\}_{t=0}^{T_{\text{pred}}}$, re-encode steps $\mathcal{R}$

**Data:** encoder $\varphi$, decoder $\varphi^{-1}$, Koopman map $K$, AFT params $W_Q, W_K, W_V$, position bias $B$, causal mask $M$

1  $z_0 \leftarrow \varphi(x_0); \quad \hat{x}_0 \leftarrow x_0; \quad \mathcal{R} \leftarrow \varnothing;$

2  **for** $t \leftarrow 1$ **to** $T_{\text{pred}}$ **do**

    *// Get original and re-encoded versions of $z_{t-1}$*

3    $z_{t-1}^{\text{orig}} \leftarrow z_{t-1}; \quad z_{t-1}^{\text{re-enc}} \leftarrow \varphi(\varphi^{-1}(z_{t-1}));$

    *// Apply AFT function to both versions*

4    build $H_{t-1} = [z_{t-T}, \dots, z_{t-1}]$ (truncate if $t < T$);

5    $\Delta z^{\text{orig}} \leftarrow \text{AFT}(z_{t-1}^{\text{orig}}, H_{t-1});$

6    $\Delta z^{\text{re-enc}} \leftarrow \text{AFT}(z_{t-1}^{\text{re-enc}}, H_{t-1});$

    *// Update both versions with their residuals*

7    $z_{t-1}^{\text{orig}} \leftarrow z_{t-1}^{\text{orig}} + \Delta z^{\text{orig}};$

8    $z_{t-1}^{\text{re-enc}} \leftarrow z_{t-1}^{\text{re-enc}} + \Delta z^{\text{re-enc}};$

    *// Apply Koopman operator to both updated versions*

9    $z_t^{\text{orig}} \leftarrow K z_{t-1}^{\text{orig}};$

10    $z_t^{\text{re-enc}} \leftarrow K z_{t-1}^{\text{re-enc}};$

    *// Calculate difference after Koopman propagation*

11    $\delta_t \leftarrow \|z_t^{\text{re-enc}} - z_t^{\text{orig}}\|_2^2;$

    *// Streaming triggers (EWMA / CUSUM / window / two-sample)*

12    **if** $\text{TriggerFires}(\delta_t; \Theta)$ **then**

13        $z_t \leftarrow z_t^{\text{re-enc}}; \quad \mathcal{R} \leftarrow \mathcal{R} \cup \{t\};$

14    **end**

15    **else**

16        $z_t \leftarrow z_t^{\text{orig}};$

17    **end**

18    $\hat{x}_t \leftarrow \varphi^{-1}(z_t);$

19  **end**

20  **return** $\{\hat{x}_t\}_{t=0}^{T_{\text{pred}}}, \mathcal{R}$

---

---

**Algorithm 3:** GRU and Transformer rollout

---

**Input:** initial window $X_0 = (x_0, x_1, \ldots, x_{T-1})$, horizon $T_{\text{pred}}$, context length $T$

**Output:** predicted states $\hat{X}_T = \{\hat{x}_t\}_{t=T}^{T+T_{\text{pred}}}$

**Data:** encoder $E$ (either $E_{gru}$ or $E_{tr}$), decoder $D_{mlp}$

1 $X_{tmp} \leftarrow X_0$;

2 **for** $t \leftarrow 1$ **to** $T_{\text{pred}}$ **do**

    *// Encode the input sequence*

3     $Z_t \leftarrow E(X_{tmp})$;

    *// Decode the latent space*

4     $\hat{x}_{t+T-1} = D_{mlp}(Z_t)$;

    *// Autoregressively prepare the next input*

5     $X_{tmp} = (x_t, x_{t+1}, \ldots, x_{t+T-1})$;

6 **end**

7 **return** $\hat{X}_T = \{\hat{x}_t\}_{t=T}^{T+T_{\text{pred}}}$

---

## G   ADDITIONAL RESULTS AND ANALYSES

### G.1   COMPLEXITY AND PARAMETER FOOTPRINT

**Attention-free latent memory vs. MHA.** Let $d$ be the latent (bottleneck) dimension and $T$ the AFT history length. The AFT block adds three linear maps $W_Q, W_K, W_V \in \mathbb{R}^{d \times d}$ and a learned position-only bias $B \in \mathbb{R}^{T \times T}$, for a total of $3d^2 + T^2$ parameters. With the settings used in most experiments ($d{=}100$, $T{=}10$), this is $\approx$ 30,100 parameters. The aggregation cost per step is linear in window length, $\mathcal{O}(Td)$, since we compute a weighted sum over the last $T$ key/value pairs with a fixed (causal) position bias (Zhai et al., 2021). By contrast, dot-product multi-head attention over a window of size $T$ requires forming attention scores over all pairs, yielding $\mathcal{O}(T^2 d)$ time and $\mathcal{O}(T^2)$ memory for the attention map, in addition to comparable linear projections.

**Total inference cost.** Per time step, the KAE backbone incurs one $d \times d$ Koopman multiply and one decode; AFT adds one extra $d \times d$ projection and a windowed $\mathcal{O}(Td)$ aggregation. The dynamic re-encoding step introduces an additional encode-decode-encode ($\varphi^{-1}$ then $\varphi$) per time step, so we incur additional $(\text{cost}[\varphi] + \text{cost}[\varphi^{-1}])$. All triggers operate in $\mathcal{O}(1)$ time per step with respect to rollout length (the two-sample test maintains fixed-size buffers, i.e., $\mathcal{O}(w)$ per update for constant $w$).

**Memory footprint.** We store the last $T$ latents ($\mathcal{O}(Td)$) and no dense $T \times T$ attention maps at inference time. This linear memory scaling enables long rollouts with a small fixed context.

### G.2   INFERENCE TIME EVALUATION

When deploying machine learning models from offline forecasting to real-time control of dynamical systems, computational efficiency becomes as critical as prediction accuracy, since control systems operate under strict timing constraints where inference delays can destabilize the entire system. We evaluated our models and inference methods on the IRMA dynamical system, predicting 100 time steps from 10 initial conditions and 5 trials per method to ensure statistical reliability of timing measurements. We report four metrics: (i) Time [s], the average wall-clock time per trial; (ii) Throughput [traj/s], the number of trajectories predicted per second; (iii) Latency [ms], the average inference time per trajectory; and (iv) Efficiency [MFLOPS], the floating-point operations executed per second, as measured using PyTorch profiler on M3 CPU hardware. These metrics reflect real executed operations rather than theoretical complexity estimates.

Table 5 presents the inference time evaluation results for all methods on the IRMA dynamical system. The Koopman-based approaches demonstrate superior computational efficiency, with Koopman AE achieving the lowest average inference time of 0.11s per trial and the highest throughput of 94.4 trajectories per second. The AFT variants, while slightly slower than the AE formulation, still offer good performance with throughput rates of 74.8-76.5 trajectories per second and notably higher computational efficiency, achieving 6774-6915 MFLOPS compared to 694 MFLOPS for the AE method. This indicates that AFT methods perform more intensive computations while maintain-

Table 5: Table of runtime performance for different models and inference methods.

| Method | Time [s] | Throughput [traj/s] | Latency [ms] | MFLOPS |
|---|---|---|---|---|
| Koopman AE | $0.11 \pm 0.16$ | 94.4 | 10.6 | 694 |
| Koopman AFT | $0.13 \pm 0.04$ | 76.5 | 13.1 | 6915 |
| Periodic AFT | $0.13 \pm 0.03$ | 74.8 | 13.4 | 6774 |
| Dyn. Reenc. AFT (Window Var) | $0.21 \pm 0.05$ | 47.6 | 21.0 | 4864 |
| Dyn. Reenc. AFT (Two Sample) | $0.38 \pm 0.08$ | 26.4 | 37.9 | 4881 |
| Transformer | $4.91 \pm 1.15$ | 2.04 | 491 | 518 |
| GRU | $6.08 \pm 2.59$ | 1.65 | 608 | 144 |

ing fast inference times. In contrast, traditional sequence models exhibit significantly higher latency, with the Transformer and GRU requiring 491ms and 608ms per trajectory, respectively.

## G.3 SEED ROBUSTNESS, PHASE-PLANE VIEWS, AND ERROR ACCUMULATION FOR THE CORE TRIO SYSTEMS

Table 6 reports mean $\pm$ std MSE across random seeds (values scaled by $\times 100$) for the Koopman baselines and our re-encoding variants. On **Duffing**, dynamic re-encoding yields the lowest error at all horizons (e.g., $1.66 \pm 0.60$ at 200 steps), reducing both mean and variance relative to KAE and AFT, and maintaining a gap through 1000 steps ($19.04 \pm 0.95$ vs. $25.63 \pm 0.97$ for KAE). This aligns with the switching-sensitive dynamics where timely snaps curb manifold drift (cf. Table 1, Fig. 2c). On the **Repressilator**, AFT without re-encoding is consistently best ($0.01 \pm 0.00$, $0.07 \pm 0.05$, $0.19 \pm 0.20$ at 200/500/1000), while snap-backs degrade performance ($\sim 0.40$–$0.71$), corroborating that triggers can inject phase resets on clean limit cycles (see §4.1). For **IRMA**, dynamic (and periodic) re-encoding dominate across horizons (e.g., $0.01 \pm 0.01$ at 200 and $0.04 \pm 0.01$ at 1000), with AFT close but consistently worse, reflecting the benefit of guarding against gradual manifold drift in higher-dimensional feedback systems.

Table 6: Mean Squared Error (MSE) over different time steps for Koopman methods running on different seed values, scaled by 100, with best values highlighted.

| Steps | Koopman AE | Koopman AFT | AFT+Re-encoding Dynamic | AFT+Re-encoding Periodic |
|---|---|---|---|---|
| *MSE over different time steps* | | | | |
| **Duffing Oscillator** | | | | |
| 200 | $11.24 \pm 0.84$ | $7.43 \pm 2.24$ | $\mathbf{1.66 \pm 0.60}$ | $1.92 \pm 0.54$ |
| 500 | $23.01 \pm 1.89$ | $22.05 \pm 3.89$ | $\mathbf{11.35 \pm 2.64}$ | $11.93 \pm 1.47$ |
| 1000 | $25.63 \pm 0.97$ | $27.54 \pm 7.48$ | $\mathbf{19.04 \pm 0.95}$ | $21.82 \pm 0.97$ |
| **Repressilator** | | | | |
| 200 | $\mathbf{0.01 \pm 0.00}$ | $\mathbf{0.01 \pm 0.00}$ | $0.40 \pm 0.05$ | $0.42 \pm 0.05$ |
| 500 | $0.23 \pm 0.05$ | $\mathbf{0.07 \pm 0.05}$ | $0.45 \pm 0.03$ | $0.35 \pm 0.01$ |
| 1000 | $1.37 \pm 0.83$ | $\mathbf{0.19 \pm 0.20}$ | $0.71 \pm 0.04$ | $0.71 \pm 0.07$ |
| **IRMA** | | | | |
| 200 | $1.18 \pm 0.20$ | $0.03 \pm 0.00$ | $\mathbf{0.01 \pm 0.01}$ | $\mathbf{0.01 \pm 0.01}$ |
| 500 | $3.13 \pm 1.31$ | $0.08 \pm 0.04$ | $\mathbf{0.02 \pm 0.02}$ | $\mathbf{0.02 \pm 0.01}$ |
| 1000 | $3.22 \pm 1.17$ | $0.12 \pm 0.03$ | $\mathbf{0.04 \pm 0.01}$ | $0.06 \pm 0.01$ |

Figure 5 complements these statistics: phase-plane/3D rollouts illustrate that re-encoding prevents rare-but-catastrophic divergence and preserves switching structure on Duffing, while remaining faithful to the attractors on Repressilator and IRMA. The MCAE curves in Fig. 6 further expose error-growth dynamics: on Duffing and IRMA, dynamic re-encoding flattens cumulative error relative to KAE and AFT, whereas on Repressilator the AFT-only curve remains lowest and most stable, consistent with Table 1 and our guidance in §5 ("When and why re-encoding helps").

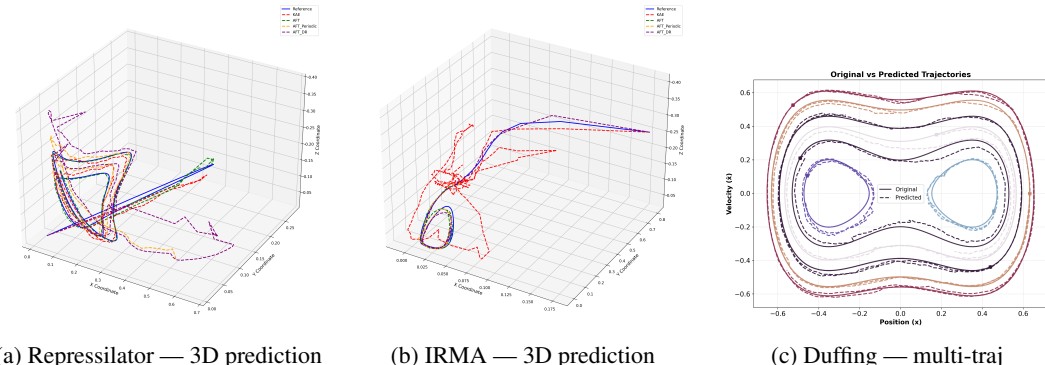

(a) Repressilator — 3D prediction     (b) IRMA — 3D prediction     (c) Duffing — multi-traj

Figure 5: **Phase Plane Visualization of the systems.** Dynamic re-encoding prevents rare-but-catastrophic divergence on long rollouts and provides robust trajectory prediction across different initial conditions.

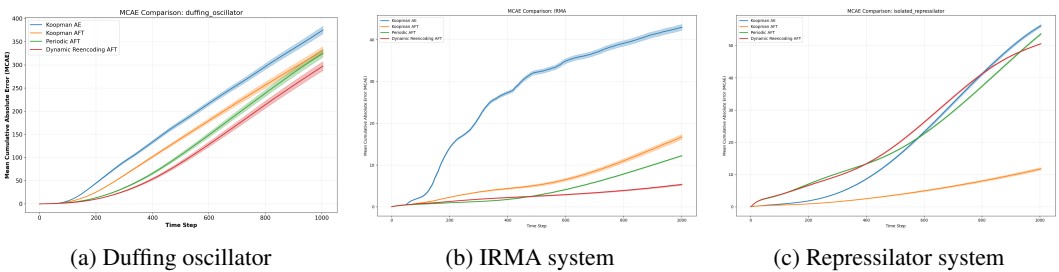

(a) Duffing oscillator     (b) IRMA system     (c) Repressilator system

Figure 6: Mean cumulative absolute error (MCAE) results for our three dynamical systems, complementing the quantitative results presented in Table 1. The plots show prediction error accumulation over time for (a) the Duffing oscillator, (b)IRMA, and (c) the Repressilator.

## G.4 ADDITIONAL DYNAMICAL SYSTEMS

To assess out-of-the-box robustness, we hold architecture and training protocols fixed across tasks (varying only loss weights) and evaluate on a diverse suite spanning continuous spectra, limit cycles, and chaos. The suite includes: the nonlinear pendulum (anharmonic, continuous spectrum), the Goodwin oscillator (sustained biochemical oscillations; complementary 200/500-step horizons), the parabolic attractor (fully linearizable by standard Koopman coordinates), the Rössler system (canonical 3D chaos), Lotka–Volterra (predator–prey oscillations), and a reduced-order fluid-flow model capturing von Kármán vortex shedding. These systems cover regimes from simple discrete spectra to chaotic attractors, providing a stringent test of generalization. Quantitative 200-step MSE results (plus 500-step for Goodwin) appear in Table 7; representative rollouts are shown in Fig. 7.

Table 7: Prediction performance comparison (MSE ↓) over 200 prediction steps across different system configurations. Lower values indicate better performance. Best results for each system are highlighted in bold.

| Model | Koopman AE | AFT | AFT with Re-encoding | |
|---|---|---|---|---|
| | – | – | Dynamic | Periodic |
| Pendulum | 0.1016 | 0.0870 | **0.0687** | 0.0695 |
| Parabolic Attractor | **0.0009** | **0.0009** | 0.0011 | 0.0010 |
| Goodwin Oscillator - 200 steps | **0.0001** | 0.0002 | 0.0026 | 0.0033 |
| Goodwin Oscillator - 500 steps | 0.0091 | **0.0009** | 0.0032 | 0.0035 |
| Lotka Volterra | 0.0112 | 0.0095 | 0.0038 | **0.0031** |
| FluidFlow | 0.0026 | 0.0019 | **0.0013** | 0.0017 |
| Rossler | 0.0085 | 0.0055 | **0.0012** | 0.0014 |

Broadly, AFT improves or matches the Koopman AE baseline, and AFT+re-encoding helps where drift accumulates (pendulum, Lotka–Volterra, fluid flow, Rössler), while offering no benefit on trivially linearizable dynamics (parabolic) or very clean short-horizon oscillations (Goodwin at 200). No per-system tuning beyond the loss weights was performed.

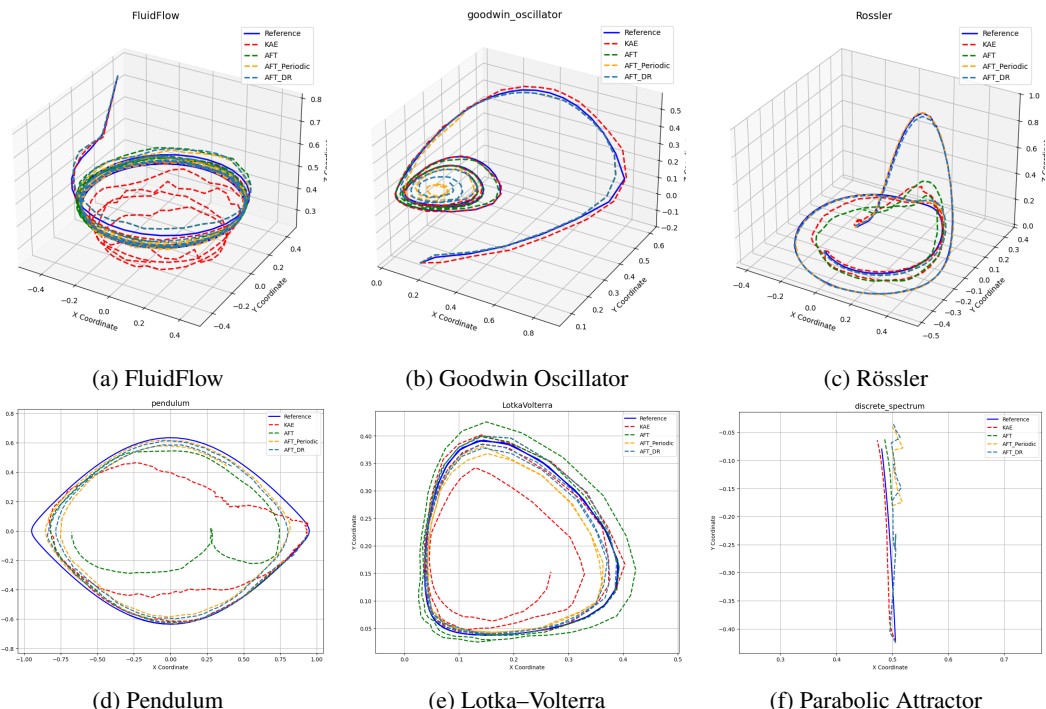

(a) FluidFlow      (b) Goodwin Oscillator      (c) Rössler

(d) Pendulum      (e) Lotka–Volterra      (f) Parabolic Attractor

Figure 7: **Additional Dynamical systems.** AFT (and AFT+Re-enc where helpful) improves or matches the baseline across diverse regimes. We did not perform additional, extensive per-system tuning.

### G.5 ABLATIONS: OPERATOR SIZE AND AFT CONTEXT

During training, re-encoding is disabled and only activated during inference (Algorithm 1). Models with larger operator sizes consistently achieve better performance than their smaller counterparts, though this performance gap narrows with the introduction of AFT. This difference is most pronounced in the Repressilator experiments (Fig 8; left). For context length, short windows ($T \in [8, 16]$) perform best (Fig 8; right). This is likely because using very long attention spans introduces memory into a system that is intended to be memoryless, and we use memory primarily for detecting drift.

## H ADDITIONAL TRAINING DETAILS AND HYPERPARAMETERS

**Rollout loss and supervision.** Given an input chunk $(x_0, \ldots, x_T)$, we encode $z_0 = \varphi(x_0)$ and roll forward with $K$ (and AFT when enabled), decoding $\hat{x}_t = \varphi^{-1}(z_t)$ at each step. We minimize the composite objective in equation 2: $\mathcal{L}_{\text{recon}}$ enforces autoencoder fidelity, $\mathcal{L}_{\text{lin}}$ encourages linear evolution $z_i \approx K^i z_0$ in latent space, $\mathcal{L}_{\text{pred}}$ supervises decoded trajectories, and $\mathcal{L}_{\text{unitary}}$ regularizes $K$. We use the full-horizon weighting (no temporal discount) to emphasize long-range accuracy.

**Optimization and schedules.** We train with AdamW (initial learning rate $10^{-3}$), a step scheduler (epochs 30/60/90, factor 0.8), batch size 128, and early stopping on validation MSE. For systems with chaotic or stiff transients, we use shorter prediction horizons during training (Table 9) for stability; inference uses the full trajectory length, and loss weights follow Table 9 ($\alpha_1, \alpha_2$ per system) and we use full-horizon weighting in equation 2 without temporal discount. The AFT block uses a causal position-only bias with a learned $T \times T$ matrix; multi-head attention baselines use identical

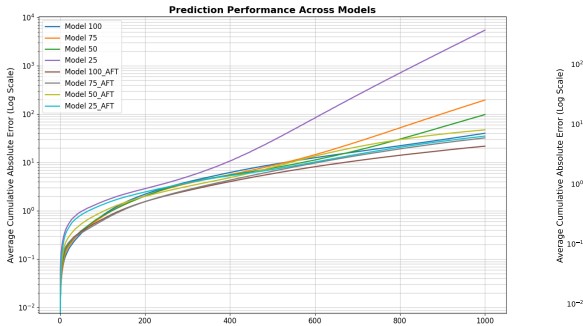 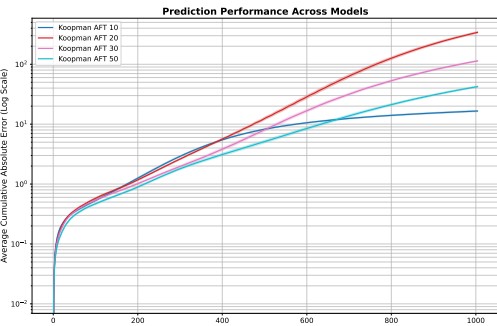

Figure 8: **Ablation studies on Koopman and AFT parameters.** Left: AFT robustness vs. Koopman with different operator sizes on Repressilator. Dense $K$ achieves the best accuracy; constrained forms need larger widths for parity. Right: AFT with different context lengths. Small context length enable learning temporal changes while longer context might lead to noise updates.

bottleneck $d$ and comparable per-head key/value sizes. Unless otherwise stated, the AFT context is $T=10$ and $K$ is dense. We report means over multiple random initial conditions; 95% CIs are shown on MCAE curves. Re-encoding is disabled during training and enabled only at inference (Alg. 2).

**Network Architecture.** We employ a symmetric autoencoder architecture with encoder and decoder networks each containing 2-4 hidden layers of equal width. We use Leaky ReLU activation functions after each hidden layer except the pre-bottleneck layer, which uses linear activation. The bottleneck dimension was initially determined from repressilator experiments and fixed at 100 dimensions across all subsequent models to ensure consistent comparison between the Koopman Autoencoder (KAE) and attention-augmented variants. This standardized architecture allows us to focus on comparing the prediction capabilities between the Koopman Autoencoder (KAE) and our attention-augmented variant.

We employed a consistent architectural framework across all dynamical systems, as detailed in Table 8. Modifications to this baseline architecture were implemented only when performance proved inadequate, with adjustments confined to operator dimensionality (bottleneck width) or the depth of hidden layers. The selection of 2–4 hidden layers was informed by preliminary experiments demonstrating that increased network depth yielded marginal performance gains while substantially elevating training instability for the dynamical systems we tested. However, this architectural choice may not generalize to dynamical systems with more complex dynamics or higher-dimensional input spaces, where deeper networks could prove beneficial.

Table 8: Architectural parameters of the models. Values are fixed unless otherwise specified.

| Parameter | Value |
|---|---|
| Bottleneck size | 100 (120 for IRMA and 128 for Rössler) |
| Autoencoder hidden layer width | 100 (128 for Rössler) |
| Autoencoder number of hidden layers | 2 (4 for Rössler) |
| AFT context length | 10 |
| Scheduler epochs | 30, 60, 90 |
| Optimizer | AdamW |

**Koopman Operator Forms.** We tested several variations of the Koopman operator, including dense, tridiagonal, diagonal, and Jordan forms. The dense form consistently outperformed the alternatives. This might be due to the additional constraints imposed by other forms, such as sparsity, block structure, or independence assumptions, which appear to limit representational capacity. Additionally, achieving complete feature disentanglement requires a larger operator size. The dense form provides maximum representational flexibility, which motivated its use throughout our experiments.

**Data Pipeline.** We divided the data into 80% training, 10% validation, and 10% testing. Model inputs for training consist of either complete trajectories or trajectory chunks, where the chunk length equals the prediction horizon, as shown in Figure 9.

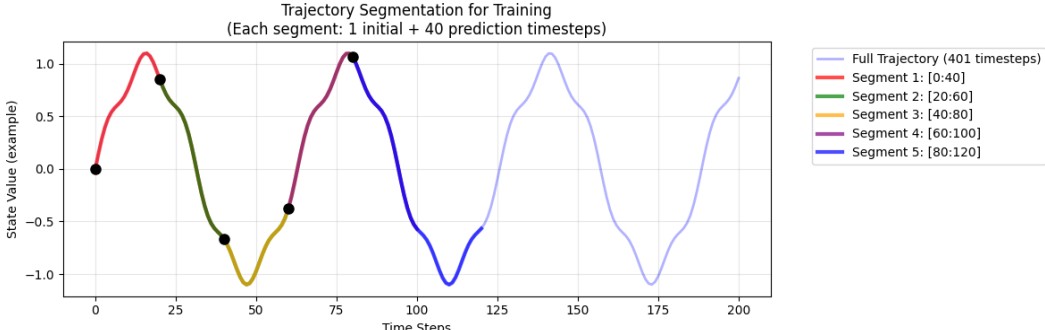

Figure 9: Trajectory segmentation for training

For complex dynamical systems exhibiting chaotic behavior, switching dynamics, or continuous spectra, we employ shorter prediction lengths during training, as this approach yields better performance and more stable training dynamics. The model unrolls predictions from the initial condition $x_0$ across the specified prediction horizon, computing both latent space predictions and their corresponding observation space reconstructions for loss evaluation. For GRU and Transformer training, the models require contextual information to learn effectively; therefore, instead of using only the initial condition $X_0$ to predict $X_1, \ldots, X_T$, we use $X_0, \ldots, X_c$ to predict $X_{c+1}, \ldots, X_T$. System-specific dataset and training settings—including sampling interval $\Delta t$, number of trajectories, $T_{\text{pred}}$, and total trajectory length—are summarized in Table 9.

Table 9: System-specific training and dataset parameters. Learning rate is fixed at $1 \times 10^{-3}$.

| System | $\alpha_1$ | $\alpha_2$ | $\Delta t$ | # Trajectories | Pred. length | Traj. length |
|---|---|---|---|---|---|---|
| Pendulum | 0.1 | 10 | 0.2 | 6000 | 40 | 200 |
| Isolated repress. | 1 | 10 | 1.25 | 15000 | 200 | 200 |
| Duffing oscillator | 0.01 | 10 | 0.05 | 6000 | 50 | 200 |
| Goodwin oscillator | 0.1 | 10 | 0.2 | 6000 | 200 | 200 |
| Lotka–Volterra | 0.01 | 10 | 1 | 6000 | 50 | 200 |
| IRMA | 2.5 | 7.4 | 2 | 3000 | 40 | 400 |
| Rössler | 0.1 | 10 | 0.05 | 2000 | 30 | 1000 |
| Fluid flow | 0.01 | 10 | 0.2 | 6000 | 50 | 200 |

