# OpenReview forum: "Learning the Koopman Operator using Attention Free Transformers"
_ICLR.cc/2026/Conference — Submitted to ICLR 2026_

### Official Review · Reviewer_Wmez · 2025-10-28

**Soundness:** 3
**Presentation:** 3
**Contribution:** 2
**Rating:** 2
**Confidence:** 5

**Summary:**

This paper addresses the critical challenge of error accumulation in long-horizon predictions using Koopman Autoencoders (KAE). The authors state that standard KAEs, while effective for single-step prediction, often suffer from phase and amplitude drift during extended rollouts, particularly in systems with continuous spectra, switching dynamics, or strong transients. This issue was identified in previous works and authors try to address it by introducing architectural innovations. Namely, they propose two complementary, lightweight components: 1) An Attention-Free Latent Memory (AFT) block that uses a short history of past latent states to compute a corrective residual before each Koopman update, reducing local errors with linear time complexity. 2) A Dynamic Re-encoding mechanism that employs online change-point detection triggers (e.g., EWMA, CUSUM, sequential two-sample tests) to detect when the latent state has drifted from the autoencoder's manifold and projects it back via an encode-decode-encode step. Importantly, the "Re-encoding" procedure, which is a purely inference time technique, is not introduced for the first time in this work, however the prior establishment of the method is NOT acknowledged throughout the paper -- refer to weaknesses section.

Through extensive experiments on three primary benchmark systems (Duffing oscillator, Repressilator, IRMA) and several additional systems, the paper demonstrates that the proposed KAE+AFT model, especially when coupled with dynamic re-encoding, achieves significantly lower long-horizon prediction error and error accumulation compared to baseline KAEs, matched-capacity Multi-Head Attention, GRU, and Transformer autoencoders, while maintaining low inference latency.

**Strengths:**

* While the core concept of re-encoding is from prior work (see Weaknesses section), the novel integration of streaming change-point detection algorithms (CUSUM, EWMA, sequential two-sample tests) to dynamically trigger the re-encoding projection is an important and pragmatic contribution. This moves the field beyond a simple fixed-period schedule to an adaptive, data-driven strategy. Furthermore, the application of a linear-complexity Attention-Free Transformer (AFT) to model local temporal context within the Koopman latent space is novel.

* The presentation is clear. The methodology is well-explained, the figures are informative, and the writing is fluent. The authors provide a detailed reproducibility checklist.

**Weaknesses:**

## Inadequate Attribution for Re-encoding Core Idea

The most substantial weakness is the presentation of the "re-encoding" mechanism as a novel contribution of this work, when in fact, _**the core concept was previously established**_. The paper _"Course Correcting Koopman Representations"_ (Fathi et al., 2023), which is cited in the bibliography _only in the context of the test environments_, introduced the fundamental idea of projecting predictions back onto the autoencoder manifold during rollout to correct drift. In fact, the motivation and very issue that paper is trying to address, was highlighted in this prior work, which again is not acknowledged in the paper at all. The only addition to "re-encoding" from this paper is to make it "dynamic" instead of "periodic" which was also alluded to in that paper.

The current manuscript, frames the introduction of re-encoding as its own. For example:

> In the Abstract: "Second, we propose dynamic re-encoding: lightweight, online change-point triggers... that detect latent drift and project predictions back onto the autoencoder manifold." The phrasing "we propose" is used without immediately clarifying that the projection itself is the prior idea, and the dynamic triggering is the novelty.

> In Section 1, Introduction: "We augment a standard KAE with two complementary pieces. ... (ii) Dynamic re-encoding uses lightweight streaming triggers... to detect latent drift and apply an encode-decode-encode (E-D-E) projection..." This again presents the entire concept (E-D-E projection + triggers) as a monolithic contribution of this paper.

> In Section 2.3, which is titled "Dynamic Re-Encoding (E-D-E projection and streaming triggers)", the projection $\mathcal{P}(z)=\phi(\phi^{−1}(z))$ is defined without citing the prior work that introduced this exact mechanism for drift correction.

More examples exist in the paper. In addition to all this, the objective (losses) and the framework they use for training is also outlined in the original re-encoding work, and again a proper acknowledgement is missing.

I cannot accept this paper without a complete overhaul of the main text to properly acknowledge and cite the original work.

---

## Incomplete Baseline for Re-encoding Ablation

A critical comparison is missing to properly contextualize the contribution of the proposed AFT module versus the dynamic triggering. The authors do not report results for the fundamental baseline: KAE with periodic re-encoding but without AFT (i.e., the core method from Fathi et al., 2023).

This omission is significant because it prevents a clear answer to a pivotal question: How much of the performance gain is due to the "dynamic" triggering, and how much is simply due to the addition of any form of re-encoding (periodic or dynamic) on top of the AFT block? It is plausible that a KAE augmented with AFT and no re-encoding at all (as shown in the paper) already outperforms a vanilla KAE with periodic re-encoding. Or a simple KAE with periodic re-encoding would outperform the same model with dynamic reencoding. If this is the case, the relative contribution of the dynamic trigger is diminished. By only showing periodic re-encoding coupled with their AFT block, the authors present an incomplete ablation that may overstate the necessity of their dynamic triggering mechanism. Including the "KAE + Periodic Re-enc (no AFT)" baseline is essential to disentangle the effects of the two proposed components and to demonstrate that the dynamic triggers provide a clear improvement over the established periodic method from prior work.

---

## Suboptimal Drift Proxy Metric

The paper uses the **Euclidean distance** in the latent space as the sole proxy for manifold drift to trigger re-encoding. This choice is not critically discussed. The Euclidean distance may be a poor metric in a learned, potentially curved latent space where the geometry of the autoencoder's manifold is complex. A more semantically meaningful distance, such as the reconstruction error in the data space or a distance computed within the feature space of the decoder itself, could be a more robust indicator of when a latent state has become "stale." The reliance on an unvalidated latent-space distance is a limitation, and the sensitivity of the trigger performance to this specific choice remains unexplored.

---

## Error Accumulating by Design

The paper introduces the AFT block to mitigate error accumulation by conditioning the Koopman update on a window of past latent states $H_t$. However, this very mechanism introduces a potential source of instability. If the latent states $z_{t-T}, ..., z_{t-1}$ already contain small errors (which is inevitable in long rollouts), the AFT block learns to produce a residual $\Delta z_t$ based on these imperfect inputs. This residual is then added to the current state and propagated by $K$, effectively injecting the history of past errors into the future trajectory. The paper does not provide a theoretical or empirical justification for why this aggregation of past states should consistently have a corrective, stabilizing effect rather than a compounding, destabilizing one. It is plausible that for certain dynamics, this "memory" could amplify high-frequency noise or reinforce systematic biases. An analysis demonstrating that the learned AFT function indeed acts as a denoiser or a phase-locking mechanism, rather than a source of error propagation, is missing and crucial for validating the core idea.

**Questions:**

* In Figure 2 and Figure 5, the multi-trajectory rollouts for systems like the Duffing oscillator appear to show significant phase drift and shape distortion, even for the proposed methods, when compared to the high-fidelity, long-term stable rollouts visually demonstrated in prior Koopman literature. Could you comment on this apparent discrepancy? Specifically, are these visual results representative of the best-case predictions from your model, or are they chosen to illustrate a challenging initial condition? Is the remaining visual drift a limitation of the learned manifold, the AFT correction, or an inherent challenge of the benchmark parameters used?

---

### Official Review · Reviewer_eheN · 2025-11-01

**Soundness:** 3
**Presentation:** 3
**Contribution:** 2
**Rating:** 4
**Confidence:** 4

**Summary:**

This paper addresses long-horizon drift in Koopman autoencoder (KAE) predictions by introducing two complementary mechanisms: (1) an attention-free transformer (AFT) block that aggregates a short window of past latents to produce corrective residuals before each Koopman update, achieving linear complexity, and (2) dynamic re-encoding using streaming change-point detection (EWMA, CUSUM, sequential two-sample tests) to project predictions back onto the learned manifold when drift is detected. The method is evaluated on three primary dynamical systems (Duffing oscillator, Repressilator, IRMA) with some additional benchmarks in the supplementary, demonstrating reduced error accumulation compared to baseline KAE and multi-head attention alternatives. The authors explicitly acknowledge the lack of theoretical guarantees and that re-encoding is inference-only.
The paper is a clean, practical engineering package with transparent costs and solid synthetic evidence, but from an ICLR perspective it lacks theoretical grounding, the re-encoding idea is incremental, and empirical validation is missing in the most compelling regimes (controlled/real-world long-horizon). As written, it’s a useful recipe, but not yet a strong conceptual advancement.

**Strengths:**

* The paper addresses a known limitation of Koopman autoencoders : phase slippage and amplitude drift over long horizons. They proposes orthogonal solutions. The AFT mechanism (addressing how we step) and re-encoding (addressing where we step) are conceptually well-motivated.

* The authors openly acknowledge failures (re-encoding degrading Repressilator performance by >10×), explicitly state the absence of theoretical guarantees, and provide comprehensive computational cost analysis.

* The paper includes extensive ablations (operator size, AFT context length, trigger families), computational efficiency analysis (Table 5), seed robustness (Table 6), and multiple dynamical systems spanning different characteristics.

* AFT achieves O(Td) complexity versus O(T²d) for multi-head attention while consistently outperforming matched-capacity MHA (Table 2). The practical throughput measurements (Table 5) validate the theoretical advantage.

*  Comprehensive appendices with all hyperparameters, data generation procedures, architectures, and code availability support reproducibility.

**Weaknesses:**

**Lacking Theoretical Justification**:
* The paper explicitly provides no convergence guarantees, spectral error bounds, or stability analysis. There's no formal justification for why AFT residuals should stabilize Koopman predictions.

* In the abstract authors claim "geometry preserving" but doesn't provide any mention of it in the rest of the paper.

Without theory, it's unclear when/why the method works, limiting generalizability and scientific understanding. Even a simple lemma showing conditions under which an AFT window of length T reduces one-step error propagation would substantially strengthen the contribution.

**Incremental Nature**:
* Novelty of dynamic re encoding over Fathi et al. periodic re-encoding feels incremental with only modest improvement. Fathi et al. already established periodic re-encoding for Koopman AEs to correct drift at inference. This paper’s adaptive triggers are a sensible extension, and AFT is a practical memory mechanism, but the conceptual step beyond “re-encode to stay on-manifold” is modest unless backed by stronger theory or harder tasks.
* AFT is used as-is from Zhai et al. (2021).
* Re-encoding via E-D-E projection builds on established consistency/projection ideas (Fathi et al. 2024,  Nayak et al. 2025, Frion et al. 2025, Noack et al. 2015) and not much contrast and comparison is provided.
* Change-point detection uses standard statistical methods (Roberts 2000, Moustakides 1986, Ross & Adams 2012) that doesn't share any relation in particular to koopman operator literature.
The contribution is essentially combining these existing pieces in the Koopman setting. While useful, this feels incremental with extremely modest gains in experiments.

**Missing Critical Comparisons and Severely Limited Scope**
* (Major) No comparison to Lu et al. (2024) and Wang et al. (2022), which directly combine Koopman with attention mechanisms and are cited in related work.
* (Minor) No or limited mention of data assimilation / long horizon forecasting works on koopman even in the extended literature. Neural koopman prior frion et al. 2024; KODA, singh et al. 2024; data assimilation operator algebra, freeman et al. 2024; Koopa, Liu et al. 2023.
* Lacking benchmark (Minor) :  No controlled systems: Despite mentioning KIC, DMDc, EDMDc, and MPC as motivating applications, all experiments are on autonomous systems. No partial/noisy observations: Real-world setting where observability is limited. No real-world benchmarks: No robotics (results on these were presented in Fathi et al. 2024), physical systems, or industrial applications. All systems are low-dimensional (2-6 states).
* Concerning GRU performance: Table 1 shows GRU+context sometimes matches the proposed method (IRMA at 200 steps: both 0.0001), questioning whether the Koopman structure is even necessary for these selected benchmarks.

**Ambiguities/confusing points**
* Equation (3): z̃_{t-1} appears without clear definition. Before this it is mentioned in the caption and figure but never formally introduced.
* Equation (5): Position bias indexing w_{t,t'} is unclear, should this be relative positions?
* Algorithm 2, line 11: Why compute drift ||z^{re-enc}_t - z^{orig}_t|| after the Koopman update rather than before?
* The claim of "element-wise linear attention" doesn't match standard definitions of linear attention

**Questions:**

In addition to the weaknesses:
* How should one decide whether to enable re-encoding for a new system without access to ground truth?
* How sensitive are the streaming tests to noise and AE reconstruction bias?
* How does performance scale with system dimensionality? What happens for 20+ state systems common in robotics/fluids?
* What precisely is “matched capacity” for MHA (heads, head dim, FFN width), and how sensitive are the outcomes?

I believe the paper has the potential if revised. Particularly focusing in adding comparisons with Koopman+attention papers, better literature positioning, Adding challenging and real world benchmark results and theoretical justification.

---

### Official Review · Reviewer_DEyJ · 2025-11-01

**Soundness:** 1
**Presentation:** 3
**Contribution:** 2
**Rating:** 2
**Confidence:** 4

**Summary:**

The paper augments a standard Koopman Autoencoder (KAE) with two mechanisms targeted at long-horizon stability: (1) an attention-free latent memory (AFT) block that aggregates a short, causal window of past latents to add a corrective residual before each linear Koopman update (linear time in window length), and (2) dynamic re-encoding that detects latent drift online (EWMA, CUSUM, sequential two-sample tests) and snaps predictions back onto the autoencoder manifold via an encode-decode-encode projection. Evaluated on Duffing, Repressilator, and IRMA, the approach reduces error accumulation versus a plain KAE, matched-capacity multi-head attention, and GRU/Transformer autoencoders (both from initial conditions and with a 50-step context). Results are reported up to 1000-step horizons with ablations over trigger policies; authors emphasize improved robustness and lower inference latency than sequence models.

**Strengths:**

- Clear problem focus (long-horizon drift) and simple remedies. The AFT residual is a drop-in, linear-time latent aggregator; dynamic re-encoding bounds accumulated drift without changing the learned K. The mechanisms are orthogonal and target local-error suppression vs. drift bounding.

- Careful baselines. On the three primary systems, KAE+AFT (and AFT+re-encoding where appropriate) consistently lowers MSE/MCAE vs. KAE and matched-capacity MHA; authors also test GRU/Transformer with and without 50-step context for fairness.

- Good guidance on when to use what. Discussion connects regimes to mechanisms: AFT-only on smooth limit cycles; AFT+triggers on switching/stiff transients.

- Complexity and latency advantages. AFT scales as $O(Td)$ vs. attention’s $O(T^2d)$, and runtime Table 5 shows substantially lower latency than Transformer/GRU.

- Additional systems (pendulum, Goodwin, Lotka–Volterra, Rössler, fluid flow) and ablations over operator size/context length strengthen the empirical picture in Appendix G.4.

**Weaknesses:**

- Incremental novelty. The AFT module is lifted from “Attention-Free Transformer” (Zhai et al., 2021) and applied as a latent residual.

- Theory gap.

- Drop-in operation claim is not supported, as it is tested on KAE architecture

- The evaluation is done only on synthetic data, whereas modern methods (e.g. [4,5]) test on real data and plenty of benchmarks/baselines are available to validate the practical significance of proposed method

- Table 1 presents mixed results (Repressilator part). I believe this is due to the synthetic benchmark saturation, related to the issue I pointed out above. The errors are very small. I am not sure if we are looking at meaningful results here or it's simply random perturbations around a technique that already works well on the data.

- The practical significance of long-horizon forecasting is not very clear to me. What are the contexts in which we need 1000 sample forecasts? As an example, suppose we have daily grain forecasting problem. What is the context in which we would need a forecast 1000 days forward, like 2.5 years, at daily grain? For all practical purposes, in this scenario we will probably only need 3 point forecast (3 years forward) at yearly grain or 36 months forward at monthly grain. Similar comments apply to other scenarios in which hierarchical forecasting is used to take care of long horizon planning problems.

- Streaming tests introduce windows/thresholds and can over-trigger. Authors tune per system. The per-system hyperparamer tuning drastically reduces the generality and significance of results, in my opinion.

**Questions:**

- Could you please explain why AFT is claimed as a contribution (first bullet point) if this is not an original contribution of this paper?

- It is not clear how the application of attention and recoding couple into Koopman theory. Equations (3) and (6) make me wonder if the resulting system preserves the basic properties of Koopman theory, i.e. are transitions still linear in the state? The notation in Figure 1(c) and equations (3-6) do not match, which further adds weight to this concern.

- The technique is advertised as a drop-in method, however it is only tested on KAE-style Koopman architecture, whereas there are multiple approaches are available. In order to support the claim of drop-in operation, experiments with additional instances of Koopman approach would be desired. Does "drop-in" generalize beyond the KAE lineage: EDMD/KDMD [1,2], HAVOC [3]? Moreover, there is more recent work, like Koopa [4] and SKOLR [5]. How would the proposed method fit into that?

- Could you please provide experiments on real data (see [4,5] for benchmarks and baselines)?

- Any results under observation noise/partial observation?

- Clarify how windows/thresholds were chosen, whether they transfer across seeds/initial conditions ranges, and report sensitivity (false positives/negatives). Can you provide transfer experiments (same thresholds across regimes) and false-trigger rates?


[1] EDMD https://arxiv.org/abs/1408.4408 \
[2] KDMD https://arxiv.org/pdf/1411.2260 \
[3] HAVOC https://arxiv.org/pdf/1608.05306 \
[4] Koopa https://arxiv.org/pdf/2305.18803 \
[5] SKOLR https://arxiv.org/pdf/2506.14113

---

> ### Comment · Reviewer_DEyJ · 2025-11-28
> **No author response**
>
> The authors provided no response. I keep my score

---

### Meta-Review · Area_Chair_je1U · 2025-12-30

**Summary:**

The paper aims to provide a solution to the long rollout drift problem for Koopman Autoencoders by augmented them with an attention free latent memory. The reviewers highlighted several issues with the paper in terms of its contributions, novelty, theory and empirical performance.

In particular, the reviewers highlighted the fact that 2 of the key claims of the paper, i.e., the AFT construction (attention-free memory) and the re-encoding mechanism were already presented, in the form used in this article, in previous papers: Zhai et. al. 2021 and Fathi et. al, 2023, respectively. This undermines the key claims of novelty in this paper.

Moreover, the authors did not provide adequate theoretical justification for their claims. In particular, no stability or convergence guarantees were provided nor did the authors provide any explanation for why AFT Residuals would stabilize predictions.

Finally, the empirical content of the paper was deemed to be substandard. The reviewers rightly commented on the lack of proper benchmarking on real data and inadequate comparison with state of the art baselines.

The authors did not provide a rebuttal making it is impossible to recommend acceptance in current form.

**Reviewer Concerns:**

Not applicable as the authors did not provide a rebuttal

**Reviewer Scores:**

Not applicable as the authors did not provide a rebuttal

---

### Decision · Program_Chairs · 2026-01-26

Reject